# Continual Learning with Soft-Masking of Parameter-Level Gradient Flow

## Abstract

Existing research on *task incremental learning* in continual learning has primarily focused on preventing *catastrophic forgetting* (CF). Several techniques have achieved learning with no CF. However, they attain it by letting each task monopolize a sub-network in a shared network, which seriously limits knowledge transfer (KT) and causes over-consumption of the network capacity, i.e., as more tasks are learned, the performance deteriorates. The goal of this paper is threefold: (1) overcoming CF, (2) encouraging KT, and (3) tackling the capacity problem. A novel and simple technique (called SPG) is proposed that *soft-masks* (partially blocks) parameter updating in training based on the importance of each parameter to old tasks. Each task still uses the full network, i.e., no monopoly of any part of the network by any task, which enables maximum KT and reduction of capacity usage. Extensive experiments demonstrate the effectiveness of SPG in achieving all three objectives. More notably, it attains significant transfer of knowledge not only among similar tasks (with shared knowledge) but also among dissimilar tasks (with little shared knowledge) while preventing CF. [1]

## 1 Introduction

Catastrophic forgetting (CF) and knowledge transfer (KT) are two key challenges of continual learning (CL), which learns a sequence of tasks incrementally. CF refers to the phenomenon where a model loses some of its performance on previous tasks once it learns a new task. KT means that tasks may help each other to learn by sharing knowledge. This work further investigates these problems in the popular CIL paradigm of CL, called *task-incremental learning* (TIL). In TIL, each task consists of several classes of objects to be learned. Once a task is learned, its data is discarded and will not be available for later use. During testing, the task id is provided for each test sample so that the corresponding classification head of the task can be used for prediction.

Several effective approaches have been proposed for TIL that can achieve learning with little or no CF. *Parameter isolation* is perhaps the most successful one in which the system learns to mask a sub-network for each task in a shared network. HAT (Serra et al., 2018) and SupSup (Wortsman et al., 2020) are two representative systems. HAT learns neurons (not parameters) that are important for each task and in learning a new task, these important neurons to previous tasks are hard-masked or blocked to prevent updating in the backward pass. Only those *free* (unmasked) *neurons* and their parameters are trainable. Thus, as more tasks are learned, the number of free neurons left becomes fewer, making later tasks harder to learn. Further, if a neuron is masked, all the parameters feeding to it are also masked, which consumes a great deal of network capacity. While in HAT, as more and more tasks are learned, it has less and less capacity to learn new tasks, which results in gradual performance deterioration (see Section 4.3), the proposed method only soft-masks parameters and thus consumes much less network capacity. SupSup uses a different approach to learn and to fix a sub-network for each task. Since it does not learn parameters but only learn separate masks per task, it can largely avoid the reduction in capacity. However, that limits knowledge transfer. As the sub-networks for old tasks cannot be updated, these approaches can have two major shortcomings: (1) *limited knowledge transfer*, and/or (2) *over-consumption of network capacity*. CAT (Ke et al., 2020) tries to improve knowledge transfer of HAT by detecting task similarities. If the new task is found similar to some previous tasks, these tasks' masks are removed so that the new task training

---

[1] The code is contained in the Supplementary Materials.

can update the parameters of these tasks for backward pass. However, this is risky because if a dissimilar task is detected as similar, serious CF occurs, and if similar tasks are detected as dissimilar, its knowledge transfer will be limited. [2]

To tackle these problems, we propose a simple and very different approach, named "*S*oft-masking of *P*arameter-level *G*radient flow" (SPG). It is surprisingly effective and contributes in following ways:

(1). Instead of learning hard/binary attentions on neurons for each task and masking/blocking these neurons in training a new task and in testing like HAT, SPG computes an importance score for each network parameter (not neuron) to old tasks using gradients. The reason that gradients can be used in the importance computation is because gradients directly tell how a change to a specific parameter will affect the output classification and may cause CF. SPG uses the importance score of each parameter as a *soft-mask* to constrain the gradient flow in backpropagation to ensure those important parameters to old tasks have minimum changes in learning the new task to prevent forgetting of previous knowledge. To our knowledge, the soft-masking of parameters has not been done before.

(2). SPG has some resemblance to the popular regularization-based approach, e.g., EWC (Kirkpatrick et al., 2017), in that both use importance of parameters to constrain changes to important parameters of old tasks. But there is a major difference. SPG directly controls each parameter (fine-grained), but EWC controls all parameters together using a regularization term in the loss to penalize the sum of changes to all parameters in the network (rather coarse-grained). Section 4.2 shows that our soft-masking is significantly better than the regularization. We believe this is an important result.

(3). In the forward pass, no masks are applied, which encourages knowledge transfer among tasks. This is better than CAT as SPG does not need CAT's extra mechanism for task similarity comparison. Knowledge sharing and transfer in SPG are automatic. SupSup cannot do knowledge transfer.

(4). As SPG soft-masks parameters, it does not monopolize any parameters or sub-network like HAT for each task and SPG's forward pass does not use any masks. This reduces the *capacity problem*.

Section 2 shows that SPG is also very different from other TIL approaches. Experiments have been conducted with (1) similar tasks to demonstrate SPG's better ability to transfer knowledge across tasks, and (2) dissimilar tasks to show SPG's ability to overcome CF and to deal with the capacity problem. SPG is superior in both, which none of the baselines is able to achieve.

## 2 RELATED WORK

Approaches in continual learning can be grouped into three main categories. We review them below.

**Regularization-based**: This approach computes importance values of either parameters or their gradients on previous tasks, and adds a regularization in the loss to restrict changes to those important parameters to mitigate CF. EWC (Kirkpatrick et al., 2017) uses the Fisher information matrix to represent the importance of parameters and a regularization to penalize the sum of changes to all parameters. SI (Zenke et al., 2017) extends EWC to reduce the complexity in computing the penalty. Many other approaches (Li & Hoiem, 2016; Zhang et al., 2020; Ahn et al., 2019) in this category have also been proposed, but they still have difficulty to prevent CF. As discussed in the introduction section, the proposed approach SPG has some resemblance to a regularization based method EWC. But the coarse-grained approach of using regularization is significant poorer than the fine-grained soft-masking in SPG for overcoming CF as we will see in Section 4.2.

**Memory-based**: This approach introduces a small memory buffer to store data from previous tasks and replay them in learning the new task to prevent CF (Lopez-Paz & Ranzato, 2017; Chaudhry et al., 2019). Some approaches (Shin et al., 2017; Deja et al., 2021) also prepare data generators for previous tasks, and the generated pseudo-samples are used instead of real samples. Although several other approaches (Rebuffi et al., 2017; Riemer et al., 2019; Aljundi et al., 2019) have been proposed, they still suffer from CF. SPG does not save any replay data or generate pseudo-replay data.

**Parameter isolation-based**: This approach is most similar to ours SPG. It tries to learn a sub-network for each task (tasks may share parameters and neurons), which limits knowledge transfer. We have discussed HAT, SupSup, and CAT in Section 1. Many others also take similar approaches,

---

[2]For example, we have conducted an experiment on a sequence of 10 similar tasks of the dataset FE-10 (see Section 4 for the data description) and only 2 out of 10 tasks were detected as similar by CAT.

**Algorithm 1** Continual Learning in SPG.

---
1: **for** $t = 1, \cdots, T$ **do**
2:      **# Training phase of task $t$. $\mathcal{M}_t$ is the model for task $t$.**
3:      **repeat**
4:         Compute gradients $\{g_i\}$ with $\mathcal{M}_t$ using $(X_t, Y_t)$.
5:         **for all** parameters of $i$-th layer **do**
6:            $g_i' \leftarrow$ Eq. (5)
7:         Update $\mathcal{M}_t$ with the modified gradients $\{g_i'\}$.
8:      **until** $\mathcal{M}_t$ converges.
9:      **# Computing the normalized importance of parameters after training task $t$.**
10:     **for** $\tau = 1, \cdots, t$ **do**
11:        Compute a loss $\mathcal{L}^{t,\tau}$ in Eq. (2).
12:        **for all** parameters of $i$-th layer **do**
13:           $\gamma_i^{t,\tau} \leftarrow$ Eq. (1)
14:     **for all** parameters of $i$-th layer **do**
15:        $\gamma_i^t \leftarrow$ Eq. (3), $\gamma_i^{\leq t} \leftarrow$ Eq. (4)
16:     Store only $\{\gamma_i^{\leq t}\}$ for future tasks.

---

e.g., Progressive Networks (PGN) (Rusu et al., 2016) and PathNet (Fernando et al., 2017). In particular, PGN allocates a sub-network for each task in advance, and progressively concatenates previous sub-networks while freezing parameters that are allocated to previous tasks. PathNet splits each layer into multiple sub-modules, and finds the best *pathway* designated to each task. In summary, parameter isolation-based methods suffer from over-consumption of network capacity and have limited knowledge transfer ability, which the proposed method tries to address.

## 3 PROPOSED SPG

As discussed in Section 1, the current parameter isolation approaches like HAT (Serra et al., 2018) and SupSup (Wortsman et al., 2020) are very effective for overcoming CF, but they hinder knowledge transfer and/or consume too much learning capacity of the network. For such a model to improve knowledge transfer, it needs to decide which parameters can be shared and updated for a new task. That is the approach taken in CAT (Ke et al., 2020). CAT finds similar tasks and removes their masks for updating, but may find wrong similar tasks, which causes CF. Further, parameters are the atomic information units, not neurons, which HAT masks. If a neuron is masked, all parameters feeding into it are masked, which costs a huge amount of learning capacity. SPG directly soft-masks parameters based on their importance to previous tasks, which is a more flexible and uses much less learning space. Soft-masking clearly enables automatic knowledge transfer.

In SPG, the importance of a parameter to a task is computed based on its gradient. We do so because gradients of parameters directly and quantitatively reflect how much changing a parameter affects the final loss. Additionally, we normalize the gradients of the parameters within each layer as we found that gradients in different layers can have very different magnitudes, which makes it difficult to compute the relative importance reliably. The normalized importance scores are accumulated by which the corresponding gradients are reduced in the optimization step to avoid forgetting the knowledge learned from the previous tasks. The whole algorithm is described in Algorithm 1.

### 3.1 COMPUTING THE NORMALIZED IMPORTANCE OF PARAMETERS

The importance of each parameter to task $t$ is computed right after completing the training of task $t$ following these steps. Task $t$'s training data, $(X_t, Y_t)$, is given again to the trained model of task $t$, and the gradient of each parameter in $i$-th layer (i.e., each weigh or bias of each layer) is then computed and used for computing the importance of the parameter. Note that we use $\theta_i$ (a vector) to represent all parameters of the $i$-th layer. This process does not update the model parameters. The reason that the importance is computed after training of the current task has converged is as follows. Even after a model converges, some parameters can have larger gradients, which indicate that changing those parameters may take the model out of the (local) minimum leading to forgetting. On the contrary, if all parameters have similar gradients (i.e, balanced directions of gradients),

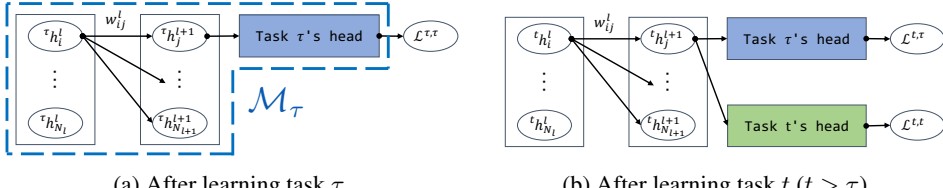

(a) After learning task $\tau$.    (b) After learning task $t$ $(t > \tau)$.

Figure 1: Cross-head importance (CHI). In the above figures, ${}^{t}h_i^l$ and $w_{ij}^l$ represents the output of the $i$-th neuron in the $l$-th layer just after training task $t$ and the parameter in $l$-th layer connecting between the neurons ${}^{t}h_i^l$ to ${}^{t}h_j^{l+1}$, respectively. (a) The importance of $w_{ij}^l$ to task $\tau$ is computed based on its gradient, $\partial\mathcal{L}^{\tau,\tau}/\partial w_{ij}^l$, normalized, and then accumulated. (b) After learning task $t$ $(t > \tau)$, the state of related parameters might have been changed. To reflect importance to task $\tau$ again with the current neurons' output (e.g., ${}^{t}h_i^l$ rather than old ${}^{\tau}h_i^l$), an additional loss, $\mathcal{L}^{t,\tau}$, is computed at task $\tau$'s head using task $t$'s data as unlabeled data for task $\tau$.

changing the parameters will not likely to change the model much to cause forgetting. Based on this assumption, we utilize the normalized gradients after training as a signal to indicate such dangerous parameter updates. In our ablation experiments, we show that taking gradients before model convergence has similar effects. The proposed mechanism in SPG has the merit that it keeps the model flexible as it does not fully block parameters using an importance threshold or binary masks. While HAT completely blocks important neurons, which results in the loss of trainable parameters over time, SPG allows most parameters remain "alive" even when most of them do not change much.

Additionally, computing the gradients based only on the model ($\mathcal{M}_t$) of the current task $t$ does not deal with another issue. We use an example illustrated in Fig. 1. For example, just after learning task $\tau$, the gradient of a parameter is computed and normalized among the same layer's parameters to be accumulated. Even though during learning task $t$ $(t > \tau)$ the parameter is not much changed considering its accumulated importance, at the end of learning task $t$, the state of related parameters might have been changed, by which the normalized importance may become less useful. To reflect the parameter's importance to task $\tau$ again in the current network state, we introduce *cross-head importance* (CHI) mechanism. In particular, an additional loss, $\mathrm{Sum}(\mathcal{M}_\tau(\boldsymbol{X}_t))$, is computed with each previous task's head by substituting task $t$'s data as unlabeled data for the previous tasks. By taking this loss, parameters affecting the logits more for previous tasks are regarded more important. Finally, both the normalized importance computed for the current task's head and the ones for previous tasks' heads in CHI are considered by taking element-wise maximum, as shown in Eq. (3).

To put things together, the proposed method computes the normalized importance, $\boldsymbol{\gamma}_i^t$, of the set of parameters of the $i$-th layer, $\boldsymbol{\theta}_i$, using each task $\tau$'s model ($1 \leq \tau \leq t$), $\mathcal{M}_\tau$, as follows:

$$\boldsymbol{\gamma}_i^{t,\tau} = \left| \tanh\left( \mathrm{Norm}\left( \frac{\partial\mathcal{L}^{t,\tau}}{\partial\boldsymbol{\theta}_i} \right) \right) \right| \tag{1}$$

$$\mathcal{L}^{t,\tau} = \begin{cases} \mathcal{L}\left( \mathcal{M}_\tau\left( \boldsymbol{X}_t \right), \boldsymbol{Y}_t \right) & (\tau = t) \\ \mathrm{Sum}\left( \mathcal{M}_\tau\left( \boldsymbol{X}_t \right) \right) & (\tau < t) \end{cases}, \quad \mathrm{Norm}(\boldsymbol{x}) = \frac{\boldsymbol{x} - \mathrm{Mean}(\boldsymbol{x})}{\sqrt{\mathrm{Var}(\boldsymbol{x}) + \epsilon}} \tag{2}$$

$$\boldsymbol{\gamma}_i^t = \max\left( \boldsymbol{\gamma}_i^{t,1}, \cdots, \boldsymbol{\gamma}_i^{t,t} \right), \tag{3}$$

where $\max(\cdot)$ means element-wise maximum, and $\epsilon$, $\mathcal{L}^{t,\tau}$ is a small value for stability and the loss function, respectively. Eq. (1) normalizes the gradients over the same layer to avoid the discrepancies caused by large differences of gradient magnitudes in different layers. For the current task's head (when $\tau = t$), a normal loss function (e.g., cross entropy) is used as $\mathcal{L}^{t,t}$ in Eq. (2). However, for each previous task's head (when $\tau < t$), since the current task data do not belong to any previous classes, the loss $\mathcal{L}^{t,\tau}$ is defined by $\mathrm{Sum}(\mathcal{M}_\tau(\boldsymbol{X}_t))$ over previous classes' logits in the proposed CHI mechanism. Essentially, this operation computes the importance of parameters based on the data of task $t$'s impact on all tasks learned so far. Finally, to prevent forgetting as much as possible, we take the accumulated importance, $\boldsymbol{\gamma}_i^{\leq t}$, as follows:

$$\boldsymbol{\gamma}_i^{\leq t} = \max\left( \boldsymbol{\gamma}_i^t, \boldsymbol{\gamma}_i^{\leq t-1} \right), \tag{4}$$

where an all-zero vector is used as $\boldsymbol{\gamma}_i^{\leq 0}$. This $\boldsymbol{\gamma}_i^{\leq t}$ depicts how important each parameter is to the learned tasks, i.e., from task $1$ to task $t$.

Table 1: Statistics of the datasets. $n$ can take 10 and 20. Validation sets are used for early stopping.

| Dataset | #Tasks | #Classes per task | #Train | #Validation | #Test |
|---------|--------|-------------------|--------|-------------|-------|
| C-$n$ | $n$ | $100/n$ | $45,000$ | $5,000$ | $10,000$ |
| T-$n$ | $n$ | $200/n$ | $90,000$ | $10,000$ | $10,000$ |
| I-100 | 100 | 10 | $1,000,000$ | $100,000$ | $50,000$ |
| FC-$n$ | $n$ | 2 | $400n$ | $40n$ | $80n$ |
| FE-$n$ | $n$ | 62 | $3100n$ | $310n$ | $620n$ |

**Memory needed to save parameter importance**: Regardless of the number of tasks, at any time only the accumulated importance $\gamma_i^{\leq t}$ is saved after the learning of each task so that it can be used again in the next task for Eq. (4). $\gamma_i^{\leq t}$ has the same size as the number of parameters, $|\boldsymbol{\theta}_i|$.

### 3.2 TRAINING PHASE

To suppress the update of important parameters in backward pass in learning task $t$, the gradients of all parameters are modified based on the accumulated importance as follow:

$$\boldsymbol{g}_i' = \left(1 - \gamma_i^{\leq t-1}\right) \boldsymbol{g}_i, \tag{5}$$

where $\boldsymbol{g}_i$ and $\boldsymbol{g}_i'$ represent the original gradients of the parameters ($\boldsymbol{\theta}_i$) of the $i$-th layer and the modified ones, which will be used in the actual optimization, respectively. Also, SPG does not employ anything special in the forward pass except it needs the task id to locate the correct classification head, which follows the standard TIL scenario. In testing, SPG behaves in the same way.

## 4 EXPERIMENTS

**Datasets**: The proposed SPG is evaluated using 5 datasets. Their statistics are given in Table 1. Below, we use "-$n$" to depict that $n$ tasks are created from each dataset ($n$ takes 10 or 20). The first three datasets are used to create dissimilar tasks to evaluate the ability of SPG in overcoming CF. The last two consist of similar tasks, which are used to evaluate SPG's knowledge transfer ability.

**(1) CIFAR100-$n$ (C-$n$)**: CIFAR100 (Krizhevsky & Hinton, 2009) is a dataset that has images of 100 classes. We split it into $n$ tasks so that each task has $100/n$ classes. **(2) TinyImageNet-$n$ (T-$n$)**: TinyImageNet (Wu et al., 2017) is a modified subset of the original ImageNet (Russakovsky et al., 2015) dataset, and has 200 classes. Each task contains $200/n$ classes. **(3) ImageNet-100 (I-100)**: ImageNet (Russakovsky et al., 2015) contains 1000 classes of objects. We split it to 100 tasks, each of which has 10 classes, to stress-test systems using a large number of tasks and classes. **(4) F-CelebA-$n$ (FC-$n$)**: Federated CelebA (Liu et al., 2015) is a dataset of celebrities' face images with several attributes. We use it with binary labels indicating whether he/she in the image is smiling or not. Each task consists of images of one celebrity. **(5) F-EMNIST-$n$ (FE-$n$)**: Federated EM-NIST (Liu et al., 2015) is a dataset that has 62 classes of hand-written symbols written by different persons. Each task consists of hand-written symbols of one person.

**Baselines**: We use 15 baselines. 10 of them are existing classical and most recent task incremental learning (TIL) systems, **EWC** (Kirkpatrick et al., 2017), **A-GEM** (Chaudhry et al., 2019), **SI** (Zenke et al., 2017), **UCL** (Ahn et al., 2019), **TAG** (Malviya et al., 2022), **PGN** (Rusu et al., 2016), **Path-Net** (Fernando et al., 2017), **HAT** (Serra et al., 2018), **CAT** (Ke et al., 2020), and **SupSup** (Wortsman et al., 2020). Additionally, three simple methods are used for references: multi-task learning (**MTL**) that trains all the tasks together, one task learning (**ONE**) that learns a separate model/network for each task and thus has no CF or KT, and naive continual learning (**NCL**) that learns each new task without taking any care of previous tasks, i.e., no mechanism to deal with CF. Since HAT, our main baseline and perhaps the most effective TIL system, adopts AlexNet (Krizhevsky et al., 2012) as its backbone, all our experiments are conducted with AlexNet. For other baselines, their original codes are used with switching their backbones to AlexNet for fair comparison. Furthermore, to compare our *soft-masking* with the *regularization-based approach* and our gradient-based importance with Fisher information matrix (FI) based importance in EWC, two more baselines **EWC-GI** and

**SPG-FI** are created. EWC-GI is EWC with its FI based importance replaced by our gradient-based importance (GI) in Section 3.1, i.e., the same penalty/regularization in EWC is applied on our accumulated importance, $\gamma_i^{\leq t-1}$ in Eq. (4) when learning task $t$ (no soft-masking). SPG-FI is SPG with our gradient-based importance replaced by FI based importance in EWC. We also examine some other baselines (e.g., hard-masking in SPG and a method replaying all data) in Appendix 6-8.

**Evaluation Metrics**: The following three metrics are used. Let $\alpha_i^j$ be the test accuracy of task $i$ task just after a model completes task $j$.

**(1)** *Accuracy*: The average of accuracy for all tasks of a dataset after learning the final task. It is computed by $1/T \sum_t^T \alpha_t^T$, where $T$ is the total number of tasks in an experiment.
**(2)** *Forward transfer*: This measures how much the learning of previous tasks contributes to the learning of the current task. It is computed by $1/T \sum_t^T (\alpha_t^t - \beta_t)$, where $\beta_t$ represents the test accuracy of task $t$ in the ONE method, which learns each task separately.
**(3)** *Backward transfer*: This measures how the learning of the current task affects the performance of the previous tasks. Negative values indicate forgetting. It is computed by $1/T \sum_t^T (\alpha_t^T - \alpha_t^t)$.

## 4.1 TRAINING DETAILS

The networks are trained with SGD by minimizing the cross-entropy loss except for TAG, which uses the RMSProp optimizer as SGD-based TAG has not been provided by the authors. The mini-batch size is 64 except MTL that uses 640 for its stability to learn more tasks and classes together. Hyper-parameters, such as the dropout rate or each method's specific hyper-paramters, are searched based on Tree-structured Parzen Estimator (Bergstra et al., 2011). With the found best hyper-parameters, the experiments are conducted 5 times with different seeds for each pair of a dataset and a method, and the average accuracy and standard deviation are reported.

## 4.2 RESULTS: ACCURACY, FORWARD TRANSFER AND BACKWARD TRANSFER

Tables 2, 3 and 4 report the accuracy, forward and backward transfer results, respectively. Each result in the tables is the average over 5 different seeds. Since CAT takes too much time to train, proportionally to the square of the number of tasks, we are unable to get its results for I-100 (ImageNet with 100 tasks) due to our limited computational resources.

**Dissimilar tasks** (C-$n$, T-$n$, I-100): MTL performs the best in all cases, but NCL performs poorly due to serious CF (negative backward transfer) as expected. While PGN, PathNet, HAT, CAT, and SupSup can achieve training with no forgetting (0 backward transfer), on average SPG clearly outperforms all of them. Although SPG slightly underperforms PathNet and SupSup in C-10 and C-20, their accuracy are markedly lower in the other settings due to PathNet's capacity problem (see Section 4.3) and SupSup's lack of knowledge sharing. The backward transfer results in Table 4 show that SPG has slight forgetting. However, its positive forward transfer results in Table 3 more than make up for the forgetting and give SPG the best final accuracy in most cases. As we explained in Section 2, the regularization-based approach is closely related to our work. However, the representative methods, EWC, SI and UCL, perform poorly (see Table 2) due to serious CF (see Table 4) that cannot be compensated by their positive forward transfer (see Table 3). Although TAG has almost no CF, its forward transfer is limited, resulting in poorer final accuracy. Comparing our gradient based importance (GI) and Fisher information (FI) matrix based importance, we observe that EWC-GI outperforms EWC except for C-20 (EWC is only 1% better), and SPG outperforms SPG-FI. Comparing soft-masking and regularization using the same importance measure, we can see SPG is markedly better than EWC-GI except for T-10 (EWC-GI is only 1% better), and SPG-FI is also better than EWC except for C-10 (EWC is only 1% better).

**Similar tasks** (FC-$n$, FE-$n$): Table 2 shows that SPG is almost as strong as MTL, and achieves the best performance among all baselines due to its positive forward (see Table 3) and positive backward knowledge transfer ability (see Table 4). NCL, A-GEM, UCL, and SI also perform well with positive or very little negative backward transfer since the tasks are similar and CF hardly happens. TAG underperforms them due to it's limited transfer. PGN, PathNet, and HAT have lower accuracy as their forward transfer is limited (i.e., they just reuse learned parameters in the forward pass) and no positive backward transfer. SupSup, which does not have any mechanism for knowledge transfer, results in much lower performance. As expected, it has no backward transfer (see Table 4). Its

Table 2: Accuracy results. Best methods in each dataset are emphasized in **bold**, and second best methods are underlined (same in the following tables).

| | Dissimilar tasks | | | | | Similar tasks | | | | |
| Model | C-10 | C-20 | T-10 | T-20 | I-100 | FC-10 | FC-20 | FE-10 | FE-20 | Avg. |
|---|---|---|---|---|---|---|---|---|---|---|
| (MTL) | $0.76_{\pm0.00}$ | $0.78_{\pm0.00}$ | $0.53_{\pm0.00}$ | $0.60_{\pm0.01}$ | $0.65_{\pm0.00}$ | $0.88_{\pm0.01}$ | $0.88_{\pm0.00}$ | $0.86_{\pm0.01}$ | $0.87_{\pm0.02}$ | 0.76 |
| (ONE) | $0.67_{\pm0.03}$ | $0.76_{\pm0.01}$ | $0.44_{\pm0.03}$ | $0.54_{\pm0.01}$ | $0.49_{\pm0.00}$ | $0.75_{\pm0.03}$ | $0.79_{\pm0.02}$ | $0.81_{\pm0.01}$ | $0.80_{\pm0.01}$ | 0.67 |
| NCL | $0.51_{\pm0.02}$ | $0.54_{\pm0.05}$ | $0.37_{\pm0.01}$ | $0.41_{\pm0.01}$ | $0.31_{\pm0.01}$ | $0.84_{\pm0.02}$ | $0.84_{\pm0.01}$ | $0.86_{\pm0.01}$ | $0.86_{\pm0.00}$ | 0.62 |
| A-GEM | $0.51_{\pm0.01}$ | $0.57_{\pm0.07}$ | $0.36_{\pm0.01}$ | $0.42_{\pm0.01}$ | $0.32_{\pm0.01}$ | $0.83_{\pm0.04}$ | $0.83_{\pm0.02}$ | $0.87_{\pm0.00}$ | $0.87_{\pm0.00}$ | 0.62 |
| PGN | $0.61_{\pm0.01}$ | $0.74_{\pm0.01}$ | $0.40_{\pm0.01}$ | $0.54_{\pm0.01}$ | $0.33_{\pm0.00}$ | $0.75_{\pm0.02}$ | $0.74_{\pm0.02}$ | $0.82_{\pm0.01}$ | $0.81_{\pm0.00}$ | 0.64 |
| PathNet | **$0.68_{\pm0.01}$** | $0.72_{\pm0.01}$ | **$0.48_{\pm0.01}$** | $0.50_{\pm0.01}$ | $0.47_{\pm0.00}$ | $0.77_{\pm0.01}$ | $0.80_{\pm0.01}$ | $0.85_{\pm0.00}$ | $0.84_{\pm0.01}$ | 0.68 |
| HAT | $0.63_{\pm0.01}$ | $0.72_{\pm0.01}$ | $0.46_{\pm0.01}$ | $0.52_{\pm0.02}$ | $0.45_{\pm0.02}$ | $0.79_{\pm0.03}$ | $0.82_{\pm0.01}$ | $0.84_{\pm0.01}$ | $0.85_{\pm0.01}$ | 0.67 |
| CAT | $0.64_{\pm0.01}$ | $0.74_{\pm0.01}$ | $0.44_{\pm0.01}$ | $0.51_{\pm0.01}$ | N/A | $0.83_{\pm0.01}$ | $0.83_{\pm0.04}$ | $0.83_{\pm0.01}$ | $0.84_{\pm0.01}$ | N/A |
| SupSup | $0.66_{\pm0.00}$ | **$0.76_{\pm0.00}$** | $0.44_{\pm0.00}$ | $0.54_{\pm0.00}$ | $0.49_{\pm0.00}$ | $0.71_{\pm0.01}$ | $0.72_{\pm0.01}$ | $0.81_{\pm0.01}$ | $0.80_{\pm0.00}$ | 0.66 |
| UCL | $0.65_{\pm0.01}$ | $0.74_{\pm0.01}$ | $0.45_{\pm0.00}$ | $0.55_{\pm0.00}$ | $0.37_{\pm0.01}$ | **$0.86_{\pm0.01}$** | **$0.87_{\pm0.01}$** | $0.85_{\pm0.01}$ | $0.85_{\pm0.02}$ | 0.69 |
| SI | $0.63_{\pm0.00}$ | $0.70_{\pm0.01}$ | $0.46_{\pm0.01}$ | $0.53_{\pm0.01}$ | $0.44_{\pm0.00}$ | **$0.86_{\pm0.01}$** | **$0.87_{\pm0.00}$** | **$0.88_{\pm0.00}$** | **$0.88_{\pm0.00}$** | 0.69 |
| TAG | $0.61_{\pm0.01}$ | $0.68_{\pm0.01}$ | $0.43_{\pm0.01}$ | $0.50_{\pm0.00}$ | $0.45_{\pm0.00}$ | $0.74_{\pm0.04}$ | $0.77_{\pm0.02}$ | $0.84_{\pm0.00}$ | $0.84_{\pm0.00}$ | 0.65 |
| EWC | $0.62_{\pm0.01}$ | $0.61_{\pm0.03}$ | $0.37_{\pm0.01}$ | $0.41_{\pm0.01}$ | $0.25_{\pm0.01}$ | $0.81_{\pm0.03}$ | $0.86_{\pm0.01}$ | $0.87_{\pm0.00}$ | $0.87_{\pm0.01}$ | 0.63 |
| EWC-GI | $0.63_{\pm0.01}$ | $0.60_{\pm0.02}$ | **$0.48_{\pm0.01}$** | $0.49_{\pm0.02}$ | $0.53_{\pm0.01}$ | $0.83_{\pm0.01}$ | $0.85_{\pm0.01}$ | $0.86_{\pm0.02}$ | $0.87_{\pm0.01}$ | 0.68 |
| SPG-FI | $0.61_{\pm0.01}$ | $0.68_{\pm0.01}$ | $0.44_{\pm0.01}$ | $0.51_{\pm0.01}$ | $0.49_{\pm0.00}$ | **$0.86_{\pm0.01}$** | **$0.87_{\pm0.01}$** | $0.86_{\pm0.01}$ | **$0.88_{\pm0.00}$** | 0.69 |
| **SPG** | $0.66_{\pm0.01}$ | $0.75_{\pm0.00}$ | $0.47_{\pm0.00}$ | **$0.57_{\pm0.00}$** | **$0.55_{\pm0.01}$** | **$0.86_{\pm0.01}$** | **$0.87_{\pm0.00}$** | $0.87_{\pm0.00}$ | $0.87_{\pm0.00}$ | **0.72** |

Table 3: Forward transfer results. MTL has no results here as it learns all tasks together.

| | Dissimilar tasks | | | | | Similar tasks | | | | |
| Model | C-10 | C-20 | T-10 | T-20 | I-100 | FC-10 | FC-20 | FE-10 | FE-20 | Avg. |
|---|---|---|---|---|---|---|---|---|---|---|
| NCL | −0.07 | −0.03 | −0.03 | −0.04 | −0.01 | +0.08 | **+0.06** | +0.04 | +0.06 | +0.01 |
| A-GEM | −0.03 | −0.01 | −0.04 | −0.04 | −0.01 | +0.08 | +0.05 | +0.04 | +0.06 | +0.01 |
| PGN | −0.06 | −0.03 | −0.04 | −0.01 | −0.16 | 0.00 | −0.04 | +0.01 | +0.01 | −0.04 |
| PathNet | +0.01 | −0.05 | +0.05 | −0.04 | −0.02 | +0.02 | +0.02 | +0.04 | +0.05 | +0.01 |
| HAT | −0.04 | −0.05 | +0.02 | −0.03 | −0.04 | +0.04 | +0.03 | +0.03 | +0.05 | 0.00 |
| CAT | −0.03 | −0.03 | 0.00 | −0.04 | N/A | +0.08 | +0.04 | +0.02 | +0.04 | N/A |
| SupSup | −0.01 | −0.01 | 0.00 | 0.00 | 0.00 | −0.04 | −0.07 | 0.00 | 0.00 | −0.01 |
| UCL | +0.04 | **+0.06** | +0.05 | +0.08 | +0.03 | +0.08 | **+0.06** | +0.03 | +0.05 | +0.05 |
| SI | **+0.07** | +0.04 | **+0.13** | **+0.09** | **+0.13** | +0.07 | **+0.06** | **+0.06** | **+0.08** | **+0.08** |
| TAG | −0.06 | −0.07 | 0.00 | −0.04 | −0.05 | −0.01 | −0.02 | +0.03 | +0.04 | −0.02 |
| EWC | 0.00 | −0.03 | −0.04 | −0.06 | −0.01 | +0.08 | **+0.06** | +0.05 | +0.07 | +0.01 |
| EWC-GI | −0.02 | +0.02 | +0.06 | +0.07 | +0.12 | **+0.09** | **+0.06** | +0.04 | +0.06 | +0.06 |
| SPG-FI | +0.01 | −0.01 | +0.05 | +0.03 | +0.10 | **+0.09** | **+0.06** | +0.04 | **+0.08** | +0.05 |
| **SPG** | +0.04 | +0.03 | +0.03 | +0.06 | +0.08 | **+0.09** | **+0.06** | +0.05 | +0.07 | +0.06 |

negative forward transfer for FC-10 and FC-20 is because it does not learn parameters as usual but masks, which is weaker as we observe. CAT is slightly better due to its stronger positive forward transfer and no negative backward transfer. We also observe that SPG-FI and EWC-GI perform similarly to SPG and EWC as suppressing updates of important parameters becomes less critical in similar tasks and thus the choice of GI or FI is not important. EWC-GI and EWC are worse than SPG and SPG-FI in FC-10 and FC-20, indicating regularization may harm the learning of new tasks.

**Summary**: SPG markedly outperforms all the baselines. When tasks are dissimilar, its positive forward transfer capability overcomes its slight forgetting (negative backward transfer) and achieves the best or competitive accuracy results. It has the strong positive forward transfer even with dissimilar tasks, which has not been realized by the other parameter isolation-based baselines. When tasks are similar, it has both positive forward and backward transfer to achieve the best accuracy results by not completely blocking parameters as parameter isolation-based methods do but only partially blocking them. Moreover, soft-masking (SPG and SPG-FI) is better than regularization (EWC-GI and EWC), and our gradient-based importance (SPG and EWC-GI) is better than FI (SPG-FI and EWC). We report the amount of compute and the network size in Tables 9 and 10 in Appendix.

## 4.3 CAPACITY CONSUMPTION

The reason that PGN, PathNet, HAT, and CAT have lower accuracy on average than SPG despite the fact that they can learn with no forgetting (see backward transfer in Table 4) is mainly because

Table 4: Backward transfer results. MTL has no results here as it learns all tasks together.

| | Dissimilar tasks | | | | | Similar tasks | | | | |
| Model | C-10 | C-20 | T-10 | T-20 | I-100 | FC-10 | FC-20 | FE-10 | FE-20 | Avg. |
|---|---|---|---|---|---|---|---|---|---|---|
| NCL | −0.09 | −0.20 | −0.03 | −0.09 | −0.18 | +0.02 | −0.01 | +0.01 | +0.01 | −0.06 |
| A-GEM | −0.13 | −0.21 | −0.04 | −0.09 | −0.16 | 0.00 | −0.01 | +0.01 | +0.01 | −0.07 |
| PGN | **0.00** | **0.00** | **0.00** | **0.00** | 0.00 | 0.00 | 0.00 | 0.00 | 0.00 | **0.00** |
| PathNet | **0.00** | **0.00** | **0.00** | **0.00** | 0.00 | 0.00 | 0.00 | 0.00 | 0.00 | **0.00** |
| HAT | **0.00** | **0.00** | **0.00** | **0.00** | 0.00 | 0.00 | 0.00 | 0.00 | 0.00 | **0.00** |
| CAT | **0.00** | **0.00** | **0.00** | **0.00** | N/A | 0.00 | 0.00 | 0.00 | 0.00 | N/A |
| SupSup | **0.00** | **0.00** | **0.00** | **0.00** | 0.00 | 0.00 | 0.00 | 0.00 | 0.00 | **0.00** |
| UCL | −0.06 | −0.09 | −0.03 | −0.07 | −0.14 | +0.03 | **+0.02** | +0.01 | +0.01 | −0.04 |
| SI | −0.12 | −0.05 | −0.11 | −0.10 | −0.17 | **+0.05** | **+0.02** | +0.01 | +0.01 | −0.05 |
| TAG | −0.01 | −0.02 | **0.00** | −0.01 | **+0.01** | 0.00 | 0.00 | 0.00 | 0.00 | **0.00** |
| EWC | −0.06 | −0.13 | −0.04 | −0.07 | −0.22 | −0.01 | **+0.02** | +0.01 | 0.00 | −0.06 |
| EWC-GI | −0.01 | −0.19 | −0.01 | −0.13 | −0.08 | 0.00 | 0.00 | **+0.02** | **+0.02** | −0.04 |
| SPG-FI | −0.07 | −0.07 | −0.04 | −0.07 | −0.10 | +0.03 | **+0.02** | **+0.02** | +0.01 | −0.03 |
| **SPG** | −0.04 | −0.04 | **0.00** | −0.03 | −0.02 | +0.02 | **+0.02** | +0.01 | +0.01 | −0.01 |

Table 5: Each cell reports how many percentage of parameters are completely blocked (i.e., importance of 1) just after learning task $t$. $T$ is the total number of tasks (e.g., $T = 10$ for C-10).

| $t$ | Model | Dissimilar tasks | | | | | Similar tasks | | | |
|---|---|---|---|---|---|---|---|---|---|---|
| | | C-10 | C-20 | T-10 | T-20 | I-100 | FC-10 | FC-20 | FE-10 | FE-20 |
| 1 | HAT | 1.91% | 15.91% | 2.88% | 0.16% | 23.50% | 0.28% | 6.29% | 23.94% | 19.03% |
| | **SPG** | 0.09% | 0.14% | 0.09% | 0.05% | 0.11% | 0.07% | 0.16% | 0.15% | 0.13% |
| $T/2$ | HAT | 22.44% | 98.24% | 36.68% | 21.06% | 99.82% | 3.40% | 65.73% | 86.81% | 93.95% |
| | **SPG** | 0.74% | 1.78% | 0.50% | 0.83% | 2.86% | 0.42% | 1.23% | 0.96% | 1.52% |
| $T$ | HAT | 44.76% | 99.61% | 57.46% | 38.64% | 99.88% | 9.76% | 79.50% | 97.86% | 97.52% |
| | **SPG** | 1.53% | 2.76% | 0.84% | 1.43% | 4.18% | 0.71% | 2.03% | 1.58% | 2.27% |

they suffer from the capacity problem, which is also indirectly reflected in lower forward transfer in Table 3. Although SupSup does not suffer from this problem, its architecture prevents transfer and gives markedly poorer performances especially in similar tasks (see Table 2). As discussed in Section 1, since these parameter isolation-based methods freeze a sub-network for each task, as more tasks are learned, the capacity of the network left for learning new knowledge becomes less and less except for SupSup, which leads to poorer performance for later tasks. Table 5 shows the percentage of parameters in the whole network that are completely blocked by HAT and SPG (layer-wise results are given in Table 7 in Appendix). It can be seen that SPG blocks much fewer parameters than HAT does in all cases. This advantage allows SPG to have more flexibility and capacity to learn while mitigating forgetting, which leads to better performances.

Fig. 2 plots the forward transfer of each task. A positive value in the figures means that a method's forward test result (the test result of the task obtained right after the task is learned) is better than ONE, benefited from the forward knowledge transfer. We can clearly observe a downward trend of these baselines for dissimilar tasks ((a)-(e)), especially (a), (b), (d), and (e). We believe that this is due to the capacity problem, i.e., as more tasks are learned, they gradually lose their learning capacity. The reason that PGN increases its performance at the beginning is because PGN splits the model into sub-networks for all tasks in advance, and when a new task comes, all previous sub-networks are combined to produce the model, which thus gradually improves the learning capability and then stabilizes. However, its performance is significantly lower than others overall. On the other hand, SPG shows a upward trend in all cases, even for dissimilar tasks, due to its positive forward transfer, which indicates SPG has a much higher capacity to learn. For Figure (f), the difference is not obvious as the tasks are similar, which can share many parameters and has less capacity issue. However, their accuracy in Table 2 are inferior to SPG due to their weaker forward transfer.

## 4.4 ABLATION STUDIES

SPG has two mechanisms that can vary. The first one is with or without *cross-head importance* (CHI), as introduced in Eq. (3), and the other is when to compute the importance of parameters. The

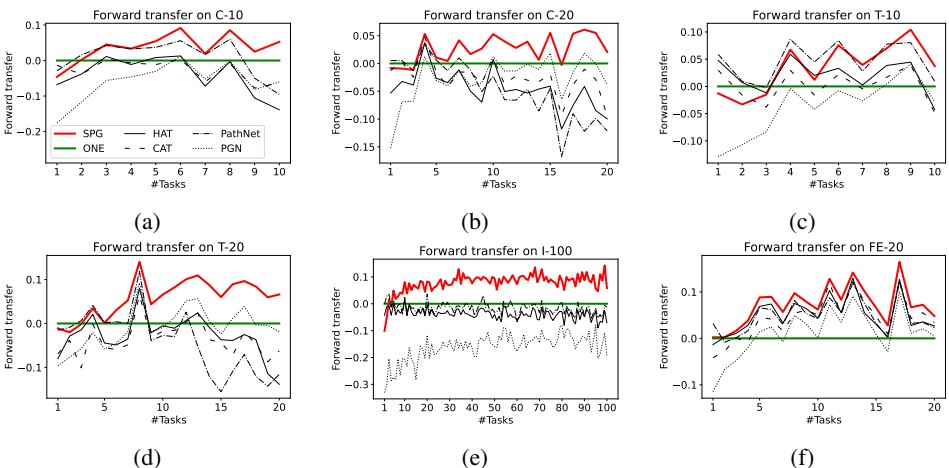

Figure 2: We plot the forward transfer of each task along with the number of tasks learned. (a) to (e) are the plots with dissimilar tasks, while (f) is the one with similar tasks. Three additional figures are given in Fig. 3 in Appendix.

Table 6: The results for the ablation studies. The averaged accuracy is presented. The averaged convergence epoch per dataset is also shown as "Cvg." for reference.

| | Dissimilar tasks | | | | | Similar tasks | | | |
|---|---|---|---|---|---|---|---|---|---|
| | C-10 | C-20 | T-10 | T-20 | I-100 | FC-10 | FC-20 | FE-10 | FE-20 |
| Cvg. | 53 | 53 | 39 | 45 | 40 | 66 | 53 | 52 | 50 |
| SPG | $\mathbf{0.66}_{\pm0.01}$ | $\mathbf{0.75}_{\pm0.00}$ | $\mathbf{0.47}_{\pm0.00}$ | $\mathbf{0.57}_{\pm0.00}$ | $\mathbf{0.55}_{\pm0.01}$ | $0.86_{\pm0.01}$ | $\mathbf{0.87}_{\pm0.00}$ | $0.87_{\pm0.00}$ | $0.87_{\pm0.00}$ |
| @e1 | $0.65_{\pm0.01}$ | $0.74_{\pm0.01}$ | $\mathbf{0.47}_{\pm0.00}$ | $0.56_{\pm0.01}$ | $\mathbf{0.55}_{\pm0.00}$ | $0.86_{\pm0.01}$ | $\mathbf{0.87}_{\pm0.00}$ | $\mathbf{0.88}_{\pm0.00}$ | $\mathbf{0.88}_{\pm0.00}$ |
| @e10 | $\mathbf{0.66}_{\pm0.01}$ | $\mathbf{0.75}_{\pm0.01}$ | $\mathbf{0.47}_{\pm0.00}$ | $0.56_{\pm0.01}$ | $\mathbf{0.55}_{\pm0.00}$ | $0.86_{\pm0.01}$ | $\mathbf{0.87}_{\pm0.01}$ | $0.87_{\pm0.00}$ | $\mathbf{0.88}_{\pm0.00}$ |
| @e20 | $0.65_{\pm0.00}$ | $0.74_{\pm0.02}$ | $\mathbf{0.47}_{\pm0.00}$ | $0.56_{\pm0.01}$ | $\mathbf{0.55}_{\pm0.01}$ | $0.86_{\pm0.01}$ | $0.86_{\pm0.01}$ | $0.87_{\pm0.00}$ | $\mathbf{0.88}_{\pm0.00}$ |
| -CHI | $0.64_{\pm0.01}$ | $0.73_{\pm0.01}$ | $\mathbf{0.47}_{\pm0.00}$ | $0.56_{\pm0.00}$ | $\mathbf{0.55}_{\pm0.01}$ | $\mathbf{0.87}_{\pm0.01}$ | $\mathbf{0.87}_{\pm0.00}$ | $\mathbf{0.88}_{\pm0.00}$ | $\mathbf{0.88}_{\pm0.00}$ |

reported results so far are based on the importance computed using gradients after model convergence. It is also possible to compute the importance in earlier epochs before model convergence.

The ablation results are presented in Table 6, where "@e$n$" means SPG employing gradients obtained after epoch $n$ (e.g., $1, 10, 20$), instead of after convergence (SPG itself), for parameter importance computation. "-CHI" means SPG without cross-head importance (CHI) where $\gamma_i^t$ is replaced with $\gamma_i^{t,t}$ in Eq. (3) without using the previous tasks' heads. We can observe from the results that the timing when the gradients are taken in importance computation has little impact on the final accuracy. It implies that the gradient direction does not change much from the beginning of training. This is perhaps not so surprising because each task does not train from scratch but based on existing network trained from old tasks and what parameters need significant changes are roughly fixed. About CHI, it improves the performance of dissimilar tasks, but may sacrifice a little in knowledge transfer in similar tasks because this mechanism blocks more gradient flow of parameters. However, even with CHI, SPG performs the best for similar tasks as shown in Table 2. Given these results, we believe that the advantage of CHI outweighs its disadvantage. More analysis of how CHI contributes to suppressing parameter updates is given in Table 8 in Appendix.

## 5 CONCLUSION

To overcome the difficulty of balancing forgetting prevention and knowledge transfer in continual learning, we proposed a novel and simple method, called SPG, that blocks/masks parameters not completely but partially to give the model more flexibility and capacity to learn. The proposed soft-masking mechanism not only overcomes CF but also performs knowledge transfer automatically. Conceptually, it is related to the regularization approach, but as we have argued and evaluated, it markedly outperforms the regularization approach. Extensive experiments have demonstrated that SPG markedly outperforms strong baselines.

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

# A  APPENDIX

## A.1  FORWARD TRANSFER

Fig. 2 in the main text of the paper shows the forward transfer plots for some datasets but not all due to space limitations. Fig. 3 here presents the forward transfer results for all datasets. It can be clearly seen that SPG has the best positive forward transfer and keeps it constantly in almost all cases. On the other hand, the other parameter isolation-based methods, PGN, PathNet, HAT, and CAT lose their ability for the forward transfer especially in later tasks for the dissimilar task experiments (i.e., (a) to (e)).

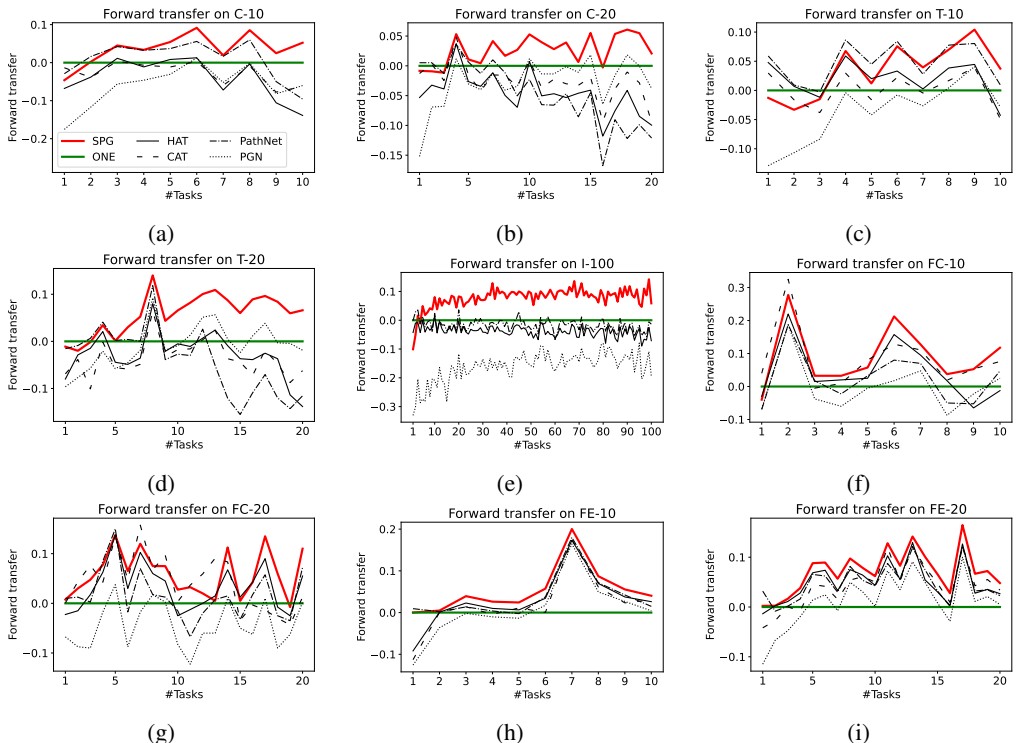

Figure 3: Forward transfer results. (a) to (e) are for dissimilar tasks, and (f) to (i) are for similar tasks.

## A.2  CAPACITY CONSUMPTION AT EACH LAYER

We show the percentage of parameters in the whole network that are fully blocked in Table 5 in the main text of the paper. Here Table 7 presents the same result for each layer. We use AlexNet as the backbone, and it has three convolution layers followed by two fully-connected layers.

As we described in Section 4.3, SPG blocks much fewer parameters than what HAT does in all cases. Additionally, we can see from Table 7 the significant difference between HAT and SPG in their layer-wise tendency. HAT blocks more parameters in earlier layers (e.g., after learning task 5 of C-10, 77% of parameters in the 1st convolution layer are completely blocked while 52% of the ones in the 2nd convolution layer are), which is reasonable given that the earlier layers are supposed to extract basic features and thus changing their parameters without being blocked could easily cause more forgetting than in later layers. On the other hand, SPG contrarily tends to completely block more parameters in later layers (e.g., after learning task 5 of C-10, 0% of parameters in the 1st convolution layer are completely blocked while 0.47% of ones in 2nd convolution layer are). Since SPG computes parameters' importance based on their gradients with regard to the loss through normalization, this result implies that later layers are likely to include more parameters on which some of the tasks highly depend. It can be said that SPG keeps earlier layers alive with less blocking

Table 7: The percentage of parameters that are completely blocked for each layer.

(a) Results for the 1st convolution layer.

| $t$ | Model | Dissimilar tasks | | | | | Similar tasks | | | |
|---|---|---|---|---|---|---|---|---|---|---|
| | | C-10 | C-20 | T-10 | T-20 | I-100 | FC-10 | FC-20 | FE-10 | FE-20 |
| 1 | HAT | 25.00% | 24.38% | 31.56% | 14.38% | 60.00% | 17.81% | 40.00% | 75.00% | 67.19% |
| | **SPG** | 0.00% | 0.00% | 0.00% | 0.00% | 0.00% | 0.00% | 0.00% | 0.00% | 0.00% |
| $T/2$ | HAT | 77.81% | 98.44% | 83.44% | 79.06% | 100.00% | 73.44% | 100.00% | 98.75% | 99.38% |
| | **SPG** | 0.00% | 0.00% | 0.00% | 0.00% | 0.11% | 0.00% | 0.00% | 0.00% | 0.00% |
| $T$ | HAT | 95.62% | 100.00% | 97.19% | 95.31% | 100.00% | 98.75% | 100.00% | 100.00% | 100.00% |
| | **SPG** | 0.00% | 0.00% | 0.00% | 0.02% | 0.22% | 0.00% | 0.00% | 0.00% | 0.00% |

(b) Results for the 2nd convolution layer.

| $t$ | Model | Dissimilar tasks | | | | | Similar tasks | | | |
|---|---|---|---|---|---|---|---|---|---|---|
| | | C-10 | C-20 | T-10 | T-20 | I-100 | FC-10 | FC-20 | FE-10 | FE-20 |
| 1 | HAT | 5.63% | 7.26% | 8.93% | 1.54% | 33.94% | 2.23% | 11.98% | 55.69% | 45.66% |
| | **SPG** | 0.07% | 0.17% | 0.02% | 0.06% | 0.03% | 0.01% | 0.02% | 0.12% | 0.09% |
| $T/2$ | HAT | 52.42% | 97.98% | 63.74% | 56.18% | 100.00% | 41.25% | 98.28% | 98.44% | 99.22% |
| | **SPG** | 0.47% | 1.92% | 0.11% | 0.32% | 0.52% | 0.03% | 0.17% | 0.59% | 0.61% |
| $T$ | HAT | 80.96% | 99.84% | 90.21% | 86.40% | 100.00% | 86.87% | 99.38% | 100.00% | 100.00% |
| | **SPG** | 0.93% | 2.54% | 0.14% | 0.37% | 0.73% | 0.05% | 0.25% | 0.84% | 0.85% |

(c) Results for the 3rd convolution layer.

| $t$ | Model | Dissimilar tasks | | | | | Similar tasks | | | |
|---|---|---|---|---|---|---|---|---|---|---|
| | | C-10 | C-20 | T-10 | T-20 | I-100 | FC-10 | FC-20 | FE-10 | FE-20 |
| 1 | HAT | 4.79% | 8.73% | 7.45% | 0.73% | 30.57% | 0.99% | 7.97% | 43.67% | 36.43% |
| | **SPG** | 0.08% | 0.19% | 0.06% | 0.04% | 0.12% | 0.04% | 0.15% | 0.09% | 0.06% |
| $T/2$ | HAT | 42.62% | 98.68% | 60.26% | 47.14% | 100.00% | 14.67% | 88.72% | 97.97% | 99.14% |
| | **SPG** | 0.61% | 2.24% | 0.27% | 0.55% | 0.54% | 0.27% | 1.12% | 0.43% | 0.43% |
| $T$ | HAT | 66.78% | 99.69% | 87.09% | 74.10% | 100.00% | 43.14% | 95.27% | 99.77% | 99.84% |
| | **SPG** | 1.14% | 2.87% | 0.36% | 0.77% | 0.66% | 0.41% | 1.62% | 0.59% | 0.59% |

(d) Results for the 4th fully-connected layer.

| $t$ | Model | Dissimilar tasks | | | | | Similar tasks | | | |
|---|---|---|---|---|---|---|---|---|---|---|
| | | C-10 | C-20 | T-10 | T-20 | I-100 | FC-10 | FC-20 | FE-10 | FE-20 |
| 1 | HAT | 2.70% | 12.59% | 3.95% | 0.21% | 25.68% | 0.34% | 6.48% | 27.72% | 22.65% |
| | **SPG** | 0.07% | 0.09% | 0.09% | 0.03% | 0.11% | 0.07% | 0.17% | 0.13% | 0.12% |
| $T/2$ | HAT | 29.71% | 98.27% | 47.66% | 29.43% | 99.88% | 3.77% | 70.90% | 90.18% | 95.64% |
| | **SPG** | 0.58% | 1.52% | 0.38% | 0.57% | 1.06% | 0.47% | 1.30% | 0.75% | 1.10% |
| $T$ | HAT | 52.36% | 99.67% | 72.28% | 50.64% | 99.91% | 12.11% | 84.04% | 98.55% | 98.20% |
| | **SPG** | 1.16% | 2.20% | 0.51% | 0.82% | 1.24% | 0.81% | 2.29% | 1.17% | 1.58% |

(e) Results for the 5th fully-connected layer.

| $t$ | Model | Dissimilar tasks | | | | | Similar tasks | | | |
|---|---|---|---|---|---|---|---|---|---|---|
| | | C-10 | C-20 | T-10 | T-20 | I-100 | FC-10 | FC-20 | FE-10 | FE-20 |
| 1 | HAT | 1.34% | 17.95% | 2.07% | 0.08% | 21.97% | 0.18% | 6.02% | 20.83% | 16.16% |
| | **SPG** | 0.10% | 0.16% | 0.09% | 0.06% | 0.11% | 0.08% | 0.16% | 0.16% | 0.14% |
| $T/2$ | HAT | 17.60% | 98.21% | 29.94% | 15.39% | 99.79% | 2.15% | 61.82% | 84.56% | 92.85% |
| | **SPG** | 0.82% | 1.90% | 0.57% | 0.97% | 3.88% | 0.40% | 1.22% | 1.08% | 1.76% |
| $T$ | HAT | 34.95% | 99.57% | 48.51% | 30.65% | 99.86% | 6.11% | 76.38% | 97.41% | 97.06% |
| | **SPG** | 1.73% | 3.05% | 1.03% | 1.77% | 5.82% | 0.68% | 1.95% | 1.82% | 2.67% |

for better basic feature learning (i.e., leading to positive knowledge transfer) while it blocks some specific parameters in later layers that are supposed to be more important for previous tasks, which is different from what HAT does. Again, we see that SPG fully blocks only a tiny percentage of parameters.

Table 8: Quantitative contribution of CHI in learning task $t$. $T$ is the total number of tasks (e.g., $T = 10$ for C-10). The average and standard deviation over 5 different seeds are presented.

| | C-10 | | | | C-20 | | | | T-10 | | | |
|---|---|---|---|---|---|---|---|---|---|---|---|---|
| $t$ | F-each | G-each | F-total | G-total | F-each | G-each | F-total | G-total | F-each | G-each | F-total | G-total |
| 2 | $0.34_{\pm 0.02}$ | $0.10_{\pm 0.01}$ | $0.23_{\pm 0.03}$ | $0.08_{\pm 0.01}$ | $0.36_{\pm 0.01}$ | $0.09_{\pm 0.01}$ | $0.25_{\pm 0.02}$ | $0.09_{\pm 0.01}$ | $0.47_{\pm 0.03}$ | $0.11_{\pm 0.00}$ | $0.30_{\pm 0.02}$ | $0.10_{\pm 0.01}$ |
| $T/2$ | $0.61_{\pm 0.03}$ | $0.22_{\pm 0.02}$ | $0.20_{\pm 0.02}$ | $0.04_{\pm 0.00}$ | $0.70_{\pm 0.03}$ | $0.27_{\pm 0.01}$ | $0.14_{\pm 0.01}$ | $0.02_{\pm 0.00}$ | $0.69_{\pm 0.03}$ | $0.23_{\pm 0.01}$ | $0.23_{\pm 0.01}$ | $0.04_{\pm 0.00}$ |
| $T$ | $0.75_{\pm 0.02}$ | $0.28_{\pm 0.01}$ | $0.13_{\pm 0.00}$ | $0.01_{\pm 0.00}$ | $0.85_{\pm 0.02}$ | $0.34_{\pm 0.01}$ | $0.06_{\pm 0.00}$ | $0.00_{\pm 0.00}$ | $0.79_{\pm 0.02}$ | $0.29_{\pm 0.01}$ | $0.14_{\pm 0.00}$ | $0.01_{\pm 0.00}$ |

| | T-20 | | | | I-100 | | | | FC-10 | | | |
|---|---|---|---|---|---|---|---|---|---|---|---|---|
| $t$ | F-each | G-each | F-total | G-total | F-each | G-each | F-total | G-total | F-each | G-each | F-total | G-total |
| 2 | $0.43_{\pm 0.02}$ | $0.11_{\pm 0.01}$ | $0.27_{\pm 0.01}$ | $0.09_{\pm 0.00}$ | $0.56_{\pm 0.03}$ | $0.12_{\pm 0.00}$ | $0.37_{\pm 0.03}$ | $0.12_{\pm 0.01}$ | $0.31_{\pm 0.04}$ | $0.08_{\pm 0.01}$ | $0.20_{\pm 0.03}$ | $0.07_{\pm 0.01}$ |
| $T/2$ | $0.78_{\pm 0.01}$ | $0.30_{\pm 0.01}$ | $0.14_{\pm 0.01}$ | $0.01_{\pm 0.00}$ | $0.89_{\pm 0.02}$ | $0.37_{\pm 0.01}$ | $0.02_{\pm 0.00}$ | $0.00_{\pm 0.00}$ | $0.49_{\pm 0.03}$ | $0.16_{\pm 0.01}$ | $0.15_{\pm 0.01}$ | $0.02_{\pm 0.00}$ |
| $T$ | $0.87_{\pm 0.00}$ | $0.35_{\pm 0.00}$ | $0.07_{\pm 0.00}$ | $0.00_{\pm 0.00}$ | $0.94_{\pm 0.01}$ | $0.43_{\pm 0.03}$ | $0.02_{\pm 0.00}$ | $0.00_{\pm 0.00}$ | $0.51_{\pm 0.04}$ | $0.18_{\pm 0.02}$ | $0.06_{\pm 0.01}$ | $0.00_{\pm 0.00}$ |

| | FC-20 | | | | FE-10 | | | | FE-20 | | | |
|---|---|---|---|---|---|---|---|---|---|---|---|---|
| $t$ | F-each | G-each | F-total | G-total | F-each | G-each | F-total | G-total | F-each | G-each | F-total | G-total |
| 2 | $0.26_{\pm 0.03}$ | $0.08_{\pm 0.01}$ | $0.20_{\pm 0.03}$ | $0.07_{\pm 0.01}$ | $0.28_{\pm 0.02}$ | $0.08_{\pm 0.01}$ | $0.20_{\pm 0.02}$ | $0.07_{\pm 0.01}$ | $0.37_{\pm 0.02}$ | $0.12_{\pm 0.01}$ | $0.25_{\pm 0.02}$ | $0.09_{\pm 0.01}$ |
| $T/2$ | $0.52_{\pm 0.05}$ | $0.19_{\pm 0.01}$ | $0.10_{\pm 0.03}$ | $0.01_{\pm 0.00}$ | $0.47_{\pm 0.02}$ | $0.17_{\pm 0.01}$ | $0.17_{\pm 0.01}$ | $0.02_{\pm 0.00}$ | $0.60_{\pm 0.01}$ | $0.24_{\pm 0.01}$ | $0.13_{\pm 0.01}$ | $0.01_{\pm 0.00}$ |
| $T$ | $0.60_{\pm 0.01}$ | $0.23_{\pm 0.01}$ | $0.07_{\pm 0.02}$ | $0.00_{\pm 0.00}$ | $0.63_{\pm 0.04}$ | $0.28_{\pm 0.02}$ | $0.14_{\pm 0.01}$ | $0.01_{\pm 0.00}$ | $0.74_{\pm 0.02}$ | $0.37_{\pm 0.01}$ | $0.10_{\pm 0.01}$ | $0.00_{\pm 0.00}$ |

## A.3 Quantitative Analysis on Cross-Head Importance (CHI)

We analyze in detail how CHI quantitatively contributes to suppressing parameter updates with the following four metrics.

**(1) Overwrite Frequency at each task (F-each)**: How often does the importance from CHI have a larger value than the one from the current task (1 means that it always happens)? It corresponds to cases where $\gamma_i^{t,\tau} > \gamma_i^{t,t}$ for any $\tau(1 \leq \tau < t)$ in Eq. (3).

**(2) Overwrite Gap at each task (G-each)**: When the cases of F-each happen, how much is the difference of overwriting on average? It is defined by the average of $\max(\gamma_i^{t,1}, \cdots, \gamma_i^{t,t-1}) - \gamma_i^{t,t}$.

**(3) Overwrite Frequency in total (F-total)**: How often does the importance from CHI actually overwrite the accumulated importance through the maximum operation? It corresponds to cases where $\gamma_i^{t,\tau} > \gamma_i^{\leq t-1}$ for any $\tau(1 \leq \tau < t)$ in Eq. (4).

**(4) Overwrite Gap in total (G-total)**: When the cases of F-total happen, how much is the difference of overwriting on average? It is defined by the average of $\max(\gamma_i^{t,1}, \cdots, \gamma_i^{t,t-1}) - \gamma_i^{\leq t-1}$.

The result is presented in Table 8. We can clearly observe that CHI adds more importance to some parameters (e.g., in C-10, about 13-23% of parameters constantly update their accumulated importance by the ones from CHI), which is denoted by F-total. Since we introduce CHI to further mitigate forgetting by accumulating more importance, this expectation is consistent with the observed results. Although CHI also overwrites the accumulated importance in similar tasks (e.g., FC-10), it overwrites less frequently (see F-each and F-total) with a smaller gap overall (see G-each and G-total), which is reasonable given that the tasks are similar.

## A.4 Computation Time

The computation time averaged over 5 different seeds is reported in Table 9. For I-100, SPG requires a significantly shorter time to learn than other parameter isolation-based approaches (i.e., PGN to SupSup). This is mainly because these approaches suffer from the capacity problem and thus need a long time to learn with limited number of free parameters, which becomes remarkable with a lot of tasks to learn (i.e., 100 tasks).

## A.5 Network Size

The number of learnable parameters of each system is presented in Table 10. Note that all approaches adopt AlexNet as their backbone, and the number of parameters vary depending on their additional structures such as attention mechanisms or sub-modules. It also depends on datasets because each dataset has a different number of tasks and in TIL, each task has a different classification head and

Table 9: Computation time for training in minutes.

| Model | Dissimilar tasks | | | | | Similar tasks | | | |
|---|---|---|---|---|---|---|---|---|---|
| | C-10 | C-20 | T-10 | T-20 | I-100 | FC-10 | FC-20 | FE-10 | FE-20 |
| (MTL) | $7_{\pm4}$ | $4_{\pm0}$ | $9_{\pm1}$ | $8_{\pm0}$ | $352_{\pm32}$ | $1_{\pm0}$ | $3_{\pm1}$ | $2_{\pm0}$ | $7_{\pm1}$ |
| (ONE) | $5_{\pm0}$ | $4_{\pm0}$ | $12_{\pm1}$ | $11_{\pm0}$ | $136_{\pm2}$ | $2_{\pm0}$ | $4_{\pm0}$ | $4_{\pm0}$ | $6_{\pm1}$ |
| NCL | $4_{\pm0}$ | $4_{\pm0}$ | $7_{\pm0}$ | $7_{\pm0}$ | $79_{\pm6}$ | $1_{\pm0}$ | $1_{\pm0}$ | $3_{\pm1}$ | $7_{\pm0}$ |
| A-GEM | $10_{\pm0}$ | $16_{\pm0}$ | $23_{\pm3}$ | $25_{\pm3}$ | $256_{\pm7}$ | $1_{\pm0}$ | $2_{\pm0}$ | $8_{\pm1}$ | $19_{\pm1}$ |
| PGN | $19_{\pm1}$ | $25_{\pm1}$ | $38_{\pm3}$ | $42_{\pm1}$ | $2918_{\pm76}$ | $2_{\pm0}$ | $5_{\pm0}$ | $15_{\pm1}$ | $45_{\pm2}$ |
| PathNet | $24_{\pm2}$ | $22_{\pm3}$ | $44_{\pm2}$ | $40_{\pm2}$ | $773_{\pm241}$ | $1_{\pm0}$ | $1_{\pm0}$ | $6_{\pm0}$ | $10_{\pm1}$ |
| UCL | $175_{\pm38}$ | $169_{\pm86}$ | $197_{\pm65}$ | $251_{\pm57}$ | $1650_{\pm66}$ | $6_{\pm0}$ | $11_{\pm1}$ | $79_{\pm27}$ | $94_{\pm7}$ |
| HAT | $14_{\pm1}$ | $14_{\pm4}$ | $22_{\pm1}$ | $29_{\pm4}$ | $1124_{\pm224}$ | $3_{\pm0}$ | $5_{\pm1}$ | $12_{\pm4}$ | $16_{\pm0}$ |
| CAT | $119_{\pm68}$ | $323_{\pm18}$ | $362_{\pm48}$ | $871_{\pm124}$ | N/A | $17_{\pm3}$ | $113_{\pm17}$ | $67_{\pm14}$ | $412_{\pm96}$ |
| SupSup | $29_{\pm6}$ | $29_{\pm6}$ | $75_{\pm0}$ | $58_{\pm1}$ | $507_{\pm5}$ | $1_{\pm0}$ | $1_{\pm0}$ | $20_{\pm0}$ | $44_{\pm1}$ |
| SI | $17_{\pm1}$ | $13_{\pm1}$ | $43_{\pm17}$ | $38_{\pm7}$ | $171_{\pm1}$ | $2_{\pm0}$ | $3_{\pm1}$ | $9_{\pm1}$ | $14_{\pm0}$ |
| TAG | $33_{\pm3}$ | $59_{\pm5}$ | $91_{\pm5}$ | $112_{\pm28}$ | $5821_{\pm244}$ | $2_{\pm0}$ | $4_{\pm1}$ | $11_{\pm1}$ | $26_{\pm4}$ |
| EWC | $15_{\pm1}$ | $16_{\pm1}$ | $19_{\pm0}$ | $27_{\pm0}$ | $375_{\pm1}$ | $1_{\pm0}$ | $3_{\pm0}$ | $5_{\pm0}$ | $16_{\pm1}$ |
| EWC-GI | $28_{\pm2}$ | $22_{\pm3}$ | $39_{\pm4}$ | $37_{\pm5}$ | $419_{\pm13}$ | $1_{\pm0}$ | $5_{\pm0}$ | $23_{\pm5}$ | $39_{\pm12}$ |
| SPG-FI | $10_{\pm1}$ | $7_{\pm0}$ | $11_{\pm0}$ | $13_{\pm5}$ | $239_{\pm7}$ | $1_{\pm0}$ | $1_{\pm0}$ | $7_{\pm0}$ | $12_{\pm3}$ |
| **SPG** | $19_{\pm4}$ | $13_{\pm2}$ | $27_{\pm4}$ | $28_{\pm1}$ | $386_{\pm30}$ | $1_{\pm0}$ | $2_{\pm0}$ | $13_{\pm0}$ | $20_{\pm0}$ |

Table 10: The number of learnable parameters of each model.

| Model | Dissimilar tasks | | | | | Similar tasks | | | |
|---|---|---|---|---|---|---|---|---|---|
| | C-10 | C-20 | T-10 | T-20 | I-100 | FC-10 | FC-20 | FE-10 | FE-20 |
| (MTL) | $6,708,772$ | $6,708,772$ | $6,913,672$ | $6,913,672$ | $8,552,872$ | $6,544,852$ | $6,585,832$ | $7,730,796$ | $9,001,176$ |
| (ONE) | $6,708,772$ | $6,708,772$ | $6,913,672$ | $6,913,672$ | $8,552,872$ | $6,544,852$ | $6,585,832$ | $7,730,796$ | $9,001,176$ |
| NCL | $6,708,772$ | $6,708,772$ | $6,913,672$ | $6,913,672$ | $8,552,872$ | $6,544,852$ | $6,585,832$ | $7,730,796$ | $9,001,176$ |
| A-GEM | $6,708,772$ | $6,708,772$ | $6,913,672$ | $6,913,672$ | $8,552,872$ | $6,544,852$ | $6,585,832$ | $7,730,796$ | $9,001,176$ |
| PGN | $1,998,653$ | $3,367,187$ | $1,365,242$ | $4,811,966$ | $797,711$ | $5,441,100$ | $4,780,766$ | $5,665,620$ | $4,750,583$ |
| PathNet | $19,503,768$ | $14,770,040$ | $22,096,648$ | $16,807,392$ | $18,497,436$ | $775,374$ | $5,198,586$ | $4,942,506$ | $2,279,946$ |
| HAT | $6,754,212$ | $6,799,652$ | $6,959,112$ | $7,004,552$ | $9,007,272$ | $6,590,292$ | $6,676,712$ | $7,776,236$ | $9,092,056$ |
| CAT | $39,525,228$ | $39,653,972$ | $40,801,912$ | $40,915,272$ | N/A | $38,531,388$ | $38,906,792$ | $46,226,660$ | $55,131,480$ |
| SupSup | $65,198,080$ | $130,191,360$ | $65,402,880$ | $130,396,160$ | $651,980,800$ | $65,034,240$ | $130,068,480$ | $65,828,480$ | $131,656,960$ |
| UCL | $6,713,316$ | $6,713,316$ | $6,918,216$ | $6,918,216$ | $8,557,416$ | $6,549,396$ | $6,590,376$ | $7,735,340$ | $9,005,720$ |
| SI | $6,708,772$ | $6,708,772$ | $6,913,672$ | $6,913,672$ | $8,552,872$ | $6,544,852$ | $6,585,832$ | $7,730,796$ | $9,001,176$ |
| TAG | $6,708,772$ | $6,708,772$ | $6,913,672$ | $6,913,672$ | $8,552,872$ | $6,544,852$ | $6,585,832$ | $7,730,796$ | $9,001,176$ |
| EWC | $6,708,772$ | $6,708,772$ | $6,913,672$ | $6,913,672$ | $8,552,872$ | $6,544,852$ | $6,585,832$ | $7,730,796$ | $9,001,176$ |
| EWC-GI | $6,708,772$ | $6,708,772$ | $6,913,672$ | $6,913,672$ | $8,552,872$ | $6,544,852$ | $6,585,832$ | $7,730,796$ | $9,001,176$ |
| SPG-FI | $6,708,772$ | $6,708,772$ | $6,913,672$ | $6,913,672$ | $8,552,872$ | $6,544,852$ | $6,585,832$ | $7,730,796$ | $9,001,176$ |
| **SPG** | $6,708,772$ | $6,708,772$ | $6,913,672$ | $6,913,672$ | $8,552,872$ | $6,544,852$ | $6,585,832$ | $7,730,796$ | $9,001,176$ |

the number of units in each classification head depends on the number of classes in each task. It can be seen that PathNet, CAT, and SupSup need more parameters than SPG and other approaches.

## A.6 COMPARISON WITH NCL

The difference between each method and NCL can be seen in Tables 2, 3 and 4 as each table gives the result of every method, including NCL. Here we only show the improvement of each method compared to NCL in terms of average accuracy, which indicates the forgetting of each system compared with NCL (Table 11). Let $\alpha_i^j$ and $\zeta_i^j$ be the test accuracy of task $i$ just after learning task $j$ by each method (as described in Section 4) and by NCL, respectively. The average accuracy difference in Table 11 is computed by $1/T \sum_t^T (\alpha_t^T - \zeta_t^T)$, where $T$ is the total number of tasks.

We can observe that SPG is the best overall (positive values indicate improvements over NCL). The best baseline is SI. Its performance is similar to SPG for similar tasks experiments as for these experiments there is a lot of knowledge sharing among tasks and thus little forgetting. However, for the dissimilar tasks experiments, SI is much worse than SPG due to much more serious forgetting.

Table 11: Accuracy difference compared to NCL.

| Model | Dissimilar tasks | | | | | Similar tasks | | | | Avg. |
|---|---|---|---|---|---|---|---|---|---|---|
| | C-10 | C-20 | T-10 | T-20 | I-100 | FC-10 | FC-20 | FE-10 | FE-20 | |
| A-GEM | 0.00 | 0.03 | −0.01 | 0.01 | 0.02 | −0.01 | −0.01 | 0.00 | 0.00 | 0.00 |
| PGN | 0.10 | 0.20 | 0.02 | 0.12 | 0.02 | −0.10 | −0.10 | −0.04 | −0.06 | 0.02 |
| PathNet | **0.17** | 0.17 | **0.11** | 0.09 | 0.17 | −0.07 | −0.04 | −0.01 | −0.02 | 0.06 |
| HAT | 0.12 | 0.18 | 0.08 | 0.11 | 0.15 | −0.05 | −0.02 | −0.02 | −0.02 | 0.06 |
| CAT | 0.13 | 0.20 | 0.06 | 0.10 | N/A | −0.01 | −0.01 | −0.03 | −0.02 | N/A |
| SupSup | 0.15 | **0.21** | 0.07 | 0.13 | 0.18 | −0.13 | −0.12 | −0.06 | −0.07 | 0.04 |
| UCL | 0.14 | 0.20 | 0.08 | 0.14 | 0.07 | **0.02** | 0.02 | −0.01 | −0.01 | 0.07 |
| SI | 0.12 | 0.16 | 0.09 | 0.11 | 0.14 | **0.02** | **0.03** | **0.01** | 0.01 | 0.08 |
| TAG | 0.10 | 0.14 | 0.06 | 0.08 | 0.14 | −0.10 | −0.07 | −0.02 | −0.03 | 0.03 |
| EWC | 0.11 | 0.06 | −0.01 | 0.00 | −0.05 | −0.03 | 0.02 | **0.01** | 0.00 | 0.01 |
| EWC-GI | 0.12 | 0.06 | **0.11** | 0.07 | 0.22 | −0.01 | 0.01 | 0.00 | 0.01 | 0.07 |
| SPG-FI | 0.10 | 0.14 | 0.07 | 0.10 | 0.18 | **0.02** | **0.03** | 0.00 | **0.02** | 0.07 |
| **SPG** | 0.15 | **0.21** | 0.10 | **0.16** | **0.25** | 0.01 | **0.03** | **0.01** | 0.01 | **0.10** |

Table 12: Accuracy results with additional baselines, B1 and B2.

| Model | Dissimilar tasks | | | | | Similar tasks | | | | Avg. |
|---|---|---|---|---|---|---|---|---|---|---|
| | C-10 | C-20 | T-10 | T-20 | I-100 | FC-10 | FC-20 | FE-10 | FE-20 | |
| (MTL) | $0.76_{\pm0.00}$ | $0.78_{\pm0.00}$ | $0.53_{\pm0.00}$ | $0.60_{\pm0.01}$ | $0.65_{\pm0.00}$ | $0.88_{\pm0.01}$ | $0.88_{\pm0.00}$ | $0.86_{\pm0.01}$ | $0.87_{\pm0.02}$ | 0.76 |
| (ONE) | $0.67_{\pm0.03}$ | $0.76_{\pm0.01}$ | $0.44_{\pm0.03}$ | $0.54_{\pm0.01}$ | $0.49_{\pm0.00}$ | $0.75_{\pm0.03}$ | $0.79_{\pm0.02}$ | $0.81_{\pm0.01}$ | $0.80_{\pm0.01}$ | 0.67 |
| NCL | $0.51_{\pm0.02}$ | $0.54_{\pm0.05}$ | $0.37_{\pm0.01}$ | $0.41_{\pm0.01}$ | $0.31_{\pm0.01}$ | $0.84_{\pm0.02}$ | $0.84_{\pm0.01}$ | $0.86_{\pm0.01}$ | $0.86_{\pm0.00}$ | 0.62 |
| B1(1.0) | $0.69_{\pm0.00}$ | $0.78_{\pm0.00}$ | $\mathbf{0.47}_{\pm0.01}$ | $0.55_{\pm0.01}$ | N/A | $\mathbf{0.86}_{\pm0.01}$ | $\mathbf{0.87}_{\pm0.01}$ | $\mathbf{0.87}_{\pm0.00}$ | $0.85_{\pm0.01}$ | N/A |
| B1(0.5) | $0.65_{\pm0.01}$ | $0.74_{\pm0.02}$ | $0.43_{\pm0.01}$ | $0.50_{\pm0.01}$ | N/A | $0.85_{\pm0.02}$ | $\mathbf{0.87}_{\pm0.00}$ | $0.85_{\pm0.00}$ | $0.85_{\pm0.00}$ | N/A |
| B1(0.1) | $0.62_{\pm0.01}$ | $0.74_{\pm0.01}$ | $0.40_{\pm0.02}$ | $0.51_{\pm0.00}$ | N/A | $0.85_{\pm0.01}$ | $\mathbf{0.87}_{\pm0.00}$ | $0.85_{\pm0.01}$ | $0.85_{\pm0.01}$ | N/A |
| B2 | $\mathbf{0.70}_{\pm0.00}$ | $\mathbf{0.79}_{\pm0.01}$ | $\mathbf{0.47}_{\pm0.00}$ | $\mathbf{0.58}_{\pm0.00}$ | $0.48_{\pm0.01}$ | $0.83_{\pm0.01}$ | $0.85_{\pm0.00}$ | $0.86_{\pm0.00}$ | $0.86_{\pm0.00}$ | 0.71 |
| **SPG** | $0.66_{\pm0.01}$ | $0.75_{\pm0.00}$ | $\mathbf{0.47}_{\pm0.00}$ | $0.57_{\pm0.00}$ | $\mathbf{0.55}_{\pm0.01}$ | $\mathbf{0.86}_{\pm0.01}$ | $\mathbf{0.87}_{\pm0.00}$ | $\mathbf{0.87}_{\pm0.00}$ | $\mathbf{0.87}_{\pm0.00}$ | **0.72** |

## A.7   COMPARISON WITH ADDITIONAL BASELINES

We have compared our SPG with existing baselines that follow the conventional continual learning (CL) setting in Section 4 in the main text. Here we additionally compare SPG with two non-conventional CL baselines suggested by an anonymous reviewer.

The two baselines are as follows: **B1**($r$): a method assuming that all data of the previous tasks are available. $r$ means the ratio of previous (replay) data used in learning a new task (e.g., when $r = 1.0$, all the previous data are used). We use $r = 1.0/0.5/0.1$. **B2**: a method that copies the network at the end of each task, and does only fine-tuning of the copy to learn the new task. The trained model is saved for the task. The results are presented in Table 12. We could not do B1 for I-100 due to our limited computational resources as it has too many tasks (100) and takes too much time.

For B1($r$), it can be seen that the larger the $r$ (1.0/0.5/0.1) is, the better the results are (except for T-20), which is obviously because it can utilize more data. What is interesting is that even B1(1.0), which uses all the previous data, can be worse than MTL in some cases (e.g., C-10, T-10, and T-20). We believe the reason is that sequentially establishing different decision boundaries for multiple tasks may sacrifice the effectiveness of learning. Although SPG is weaker than B1(1.0) in dissimilar tasks, it outperforms B1(0.5), which indicates SPG is more effective at least than storing one half of the previous data. Note that SPG does not store any previous data. For similar tasks, B1(0.5) and B(0.1) are weaker than SPG. B1(1.0) has the same results as SPG.

For B2, it enjoys forward knowledge transfer, but since B2 is not a typical continual learning method, it has no catastrophic forgetting (CF) issue. It is thus not surprising that B2 outperforms SPG on the first two dissimilar tasks (C-10 and C-20). But SPG shows competitive performance with B2 even under CF of continual learning for T-10 and T-20, and significantly outperforms B2 on I-100.

Table 13: Accuracy results with additional baselines, B1 and B2.

| Model | Dissimilar tasks | | | | |
| | T-50 | T-100 | I-50 | I-100 | I-200 |
|---|---|---|---|---|---|
| (ONE) | $0.71_{\pm 0.01}$ | $\mathbf{0.85}_{\pm 0.00}$ | $0.40_{\pm 0.00}$ | $0.49_{\pm 0.00}$ | $0.60_{\pm 0.00}$ |
| B2 | $\mathbf{0.72}_{\pm 0.00}$ | $0.84_{\pm 0.00}$ | $0.38_{\pm 0.00}$ | $0.48_{\pm 0.01}$ | $0.62_{\pm 0.00}$ |
| **SPG** | $0.71_{\pm 0.01}$ | $0.84_{\pm 0.01}$ | $\mathbf{0.45}_{\pm 0.01}$ | $\mathbf{0.55}_{\pm 0.01}$ | $\mathbf{0.66}_{\pm 0.00}$ |

Table 14: Accuracy results with HPG.

| Model | Dissimilar tasks | | | | | Similar tasks | | | | Avg. |
| | C-10 | C-20 | T-10 | T-20 | I-100 | FC-10 | FC-20 | FE-10 | FE-20 | |
|---|---|---|---|---|---|---|---|---|---|---|
| (MTL) | $0.76_{\pm 0.00}$ | $0.78_{\pm 0.00}$ | $0.53_{\pm 0.00}$ | $0.60_{\pm 0.01}$ | $0.65_{\pm 0.00}$ | $0.88_{\pm 0.01}$ | $0.88_{\pm 0.00}$ | $0.86_{\pm 0.01}$ | $0.87_{\pm 0.02}$ | 0.76 |
| (ONE) | $0.67_{\pm 0.03}$ | $0.76_{\pm 0.01}$ | $0.44_{\pm 0.03}$ | $0.54_{\pm 0.01}$ | $0.49_{\pm 0.00}$ | $0.75_{\pm 0.03}$ | $0.79_{\pm 0.02}$ | $0.81_{\pm 0.01}$ | $0.80_{\pm 0.01}$ | 0.67 |
| NCL | $0.51_{\pm 0.02}$ | $0.54_{\pm 0.05}$ | $0.37_{\pm 0.01}$ | $0.41_{\pm 0.01}$ | $0.31_{\pm 0.01}$ | $0.84_{\pm 0.02}$ | $0.84_{\pm 0.01}$ | $0.86_{\pm 0.01}$ | $0.86_{\pm 0.00}$ | 0.62 |
| HAT | $0.63_{\pm 0.01}$ | $0.72_{\pm 0.01}$ | $0.46_{\pm 0.01}$ | $0.52_{\pm 0.02}$ | $0.45_{\pm 0.02}$ | $0.79_{\pm 0.03}$ | $0.82_{\pm 0.01}$ | $0.84_{\pm 0.01}$ | $0.85_{\pm 0.01}$ | 0.67 |
| SupSup | $\mathbf{0.66}_{\pm 0.00}$ | $\mathbf{0.76}_{\pm 0.00}$ | $0.44_{\pm 0.00}$ | $0.54_{\pm 0.00}$ | $0.49_{\pm 0.00}$ | $0.71_{\pm 0.01}$ | $0.72_{\pm 0.01}$ | $0.81_{\pm 0.01}$ | $0.80_{\pm 0.00}$ | 0.66 |
| HPG | $0.63_{\pm 0.01}$ | $0.73_{\pm 0.01}$ | $0.45_{\pm 0.00}$ | $0.53_{\pm 0.01}$ | $0.49_{\pm 0.01}$ | $\mathbf{0.87}_{\pm 0.01}$ | $0.86_{\pm 0.01}$ | $\mathbf{0.87}_{\pm 0.00}$ | $\mathbf{0.87}_{\pm 0.00}$ | 0.70 |
| **SPG** | $\mathbf{0.66}_{\pm 0.01}$ | $0.75_{\pm 0.00}$ | $\mathbf{0.47}_{\pm 0.00}$ | $\mathbf{0.57}_{\pm 0.00}$ | $\mathbf{0.55}_{\pm 0.01}$ | $0.86_{\pm 0.01}$ | $\mathbf{0.87}_{\pm 0.00}$ | $\mathbf{0.87}_{\pm 0.00}$ | $\mathbf{0.87}_{\pm 0.00}$ | **0.72** |

To have a better understanding of this interesting phenomenon, we conduct additional experiments with more tasks, T-50, T-100, I-50, and I-200. We show the results in Table 13.

We can see that B2 is not better than SPG and in fact for the last three, it is actually significantly poorer than SPG. We believe the reason is as follows. The initialization of parameters affects the performance of learning of a new task, and a model that has learned a task does not always provide good initialized parameters for the next task. For the ImageNet data, which is used for I-50, I-100, and I-200, it is a more difficult dataset. In this case, the model parameters get optimized more specifically for each task, and the learning of the next task in B2 may start with parameters that are far from a good initialization, which leads to poorer results than SPG. On the other hand, since SPG preserves knowledge from all previous tasks, the accumulated knowledge in the network enables the model to utilize a good set of parameters to learn the new tasks (see below for similar tasks as well) to produce better results, which is indicated by its strong transfer compared to ONE and also as described in Table 3 in the paper.

SPG is also consistently better than B2 for similar tasks. This is because of the limited knowledge transfer in B2. Let the current task learned by B2 be $t$. B2 can only transfer knowledge from the $t-1$ task model to task $t$. Although some knowledge of earlier tasks before $t-1$ may still be in the model for $t-1$, it is largely forgotten because B2 has no mechanism to handle forgetting. Our SPG can preserve the task specific knowledge of each task and thus is able to leverage the knowledge from more and suitable tasks to learn task $t$ better.

### A.8 COMPARISON BETWEEN SOFT-MASKING AND HARD-MASKING

HAT uses hard-masking (on neurons) and we have shown that our SPG (soft-masking parameters) outperforms HAT significantly. SupSup hard-masks parameters. SPG also outperforms SupSup. Although SPG directly uses importance score (0 to 1) to mask parameters and does not have a threshold for hard-masking (0 or 1), we create a hard mask version of SPG, called "HPG" (i.e., Hard-masking, not soft-masking) by applying a threshold to convert an importance value (0 to 1) to a binary value (0 or 1). For example, if the threshold is 0.5, HPG treats importance larger than 0.5 as 1 (blocking), otherwise 0 (not blocking). We search for the best threshold for each dataset from 0.1/0.3/0.5/0.7/0.9, and report the best accuracy results in Table 14.

The results still show SPG is superior. For dissimilar tasks, HPG that does hard-masking is significantly worse than SPG, which means that it is difficult to manually control the threshold and it is more effective to use the importance scores as soft-masks (as in SPG). On the other hand, both HPG and SPG work similarly for similar tasks, which is reasonable as for similar tasks, completely

blocking parameters is not as harmful as for dissimilar tasks because similar tasks share a lot of knowledge or parameters.

