# OpenReview forum: "Continual Learning with Soft-Masking of Parameter-Level Gradient Flow"
_ICLR.cc/2023/Conference — Submitted to ICLR 2023_

### Official Review · Reviewer_7vyQ · 2022-10-23

**Confidence:** 5
**Correctness:** 4
**Technical Novelty And Significance:** 3
**Empirical Novelty And Significance:** 3
**Recommendation:** 6

**Clarity, Quality, Novelty And Reproducibility:**

good on all accounts.
The English can be improved however.

**Strength And Weaknesses:**

**Strengths**

The method is simple and sensible.
The empirical study is extensive and appropriate.
My intuition is that the reported results would generalize to most task-incremental scenarios.

**Weaknesses**

My main concern is about the studied scenario.
My opinion is that task-incremental learning isn't aligned with practical use-cases and over-studied.
I understand that it's a useful setting to study catastrophic forgetting.
However, we are seeing increasing evidence that catastrophic forgetting isn't as catastrophic as previously thought [1], especially in task-incremental scenarios where one does not have to deal with the problem of calibrating a single-head on all tasks.

Furthermore, most literature in this space often get away with omitting the replay baseline, which is challenging to outperform, at least when computational resources are correctly fixed throughout the methods.

Now, I do think that in the realm of task-incremental papers, SPG is a good one.
I'm willing to increase my score if the SPG survives the introduction of the following baselines:
- replay where all past data is allowed (memory is cheap, so there's no need for artificially constraining to small buffers).
    - try different current vs replay task balances
- this one is really important: copy the network at the end of each task, and only finetune the copy
    - this method is simple, doesn't incur forgetting, should allow for some forward transfer, and has the same computational requirements as other baselines.


[1] Pretrained Language Model in Continual Learning: A Comparative Study


**Summary Of The Paper:**

The paper introduces a novel task-incremental approach that diminishes the gradients based on their importance (SPG).
Specifically, the importance is calculated as the normalized gradients on the current and past tasks.
For the current task, the gradients are taken w.r.t. the log-likelihood of the data.
For past tasks, the logits' gradients are used instead.

An extensive empirical study shows that SPG is on average superior than task-incremental baselines.

**Summary Of The Review:**

Simple and sensible approach.

Because of the weaknesses of the studied scenarios, the authors need some particular new baselines (see **weaknesses**)

---- POST REBUTTAL ------

I updated my score from a 5 to 6 in light of the added baselines.

---

> ### Author Response · Authors · 2022-11-17
> **To Reviewer 7vyQ (1/2)**
>
> We appreciate your review. We answer your concerns as follows.
>
> [Response 1]
> > My main concern is about the studied scenario. My opinion is that task-incremental learning isn't aligned with practical use-cases and over-studied. I understand that it's a useful setting to study catastrophic forgetting. However, we are seeing increasing evidence that catastrophic forgetting isn't as catastrophic as previously thought [1], especially in task-incremental scenarios where one does not have to deal with the problem of calibrating a single-head on all tasks.
>
> We have seen a lot of applications of task-incremental learning (TIL). We humans learn a lot of separate tasks all the time, e.g., (1) classifying different kinds of birds and (2) classifying different types of animals (notice the overlap of the concepts, which is not suitable for class-incremental learning). TIL problems are also extensive in natural language processing, e.g., different text summarization tasks and different information extraction tasks. We do agree that TIL is not hard in terms of overcoming catastrophic forgetting. However, knowledge transfer and the limited capacity issue of the existing methods are very challenging to solve. Yet, they are very important problems in practice. Knowledge transfer is particularly critical because if there is no knowledge transfer in TIL, in practice, one could just build a separate model for each task. This paper deals with forgetting, limited capacity, and knowledge transfer in TIL. To our knowledge, none of the existing methods have attempted all three. As a community, we should not discriminate against TIL as it has its own wide ranges of applications and challenges, especially knowledge transfer.
>
>
>
>
> [Response 2]
> > Furthermore, most literature in this space often get away with omitting the replay baseline, which is challenging to outperform, at least when computational resources are correctly fixed throughout the methods.
>
> We believe the reason is that for TIL, catastrophic forgetting is largely a solved problem without the need of any replay data. We agree that for CIL, replay is a very effective approach. Our work focuses on TIL and particularly, the network capacity and knowledge transfer.
>
>
>
> [Response 3]
> > Now, I do think that in the realm of task-incremental papers, SPG is a good one. I'm willing to increase my score if the SPG survives the introduction of the following baselines: 1) replay where all past data is allowed, 2)  copy the network at the end of each task, and only finetune the copy
>
> We believe that your suggested baselines are not continual learning (CL) methods. For the first baseline, when the ratio is 1, we need to save all the data, which is like running multitask learning at each task. The computational cost is huge $O(n^2)$, where $n$ is the number of tasks in a dataset. Even for humans, it is hard to imagine that whenever we need to learn a new task, we must retrieve from our memory all the data of all previous tasks that we have learned in our life-time to re-learn everything. For the second baseline, it is not a CL setting because it saves a model for each task and the model is trained with the parameters learned from all previous tasks as the initialization.
>
> Nevertheless, we have conducted new experiments for the two suggested baselines. For the first baseline denoted by B1(r), we use $r=1.0/0.5/0.1$, where $r$ means the ratio of previous (replay) data used in learning a new task (e.g., when $r=1.0$, all the previous data are used). We use B2 to denote the second "copy-and-finetune" baseline. The results are presented below. We could not do B1 for I-100 due to our limited computational resources as it has too many tasks (100) and takes too much time as it needs $O(n^2)$, where $n=100$.
>
> (C-10 to I-100 are dissimilar datasets, while FC-10 to FE-20 are similar datasets)
>
> |Model|C-10|C-20|T-10|T-20|I-100|FC-10|FC-20|FE-10|FE-20|
> |--|--|--|--|--|--|--|--|--|--|
> |(MTL)|0.76|0.78|0.53|0.60|0.65|0.88|0.88|0.86|0.87|
> |(ONE)|0.67|0.76|0.44|0.54|0.49|0.75|0.79|0.81|0.80|
> |NCL|0.51|0.54|0.37|0.41|0.31|0.84|0.84|0.86|0.86|
> |B1(1.0)|0.69|0.78|0.47|0.55|-|0.86|0.87|0.87|0.85|
> |B1(0.5)|0.65|0.74|0.43|0.50|-|0.85|0.87|0.85|0.85|
> |B1(0.1)|0.62|0.74|0.40|0.51|-|0.85|0.87|0.85|0.85|
> |B2|0.70|0.79|0.47|0.58|0.48|0.83|0.85|0.86|0.86|
> |**SPG**|0.66|0.75|0.47|0.57|0.55|0.86|0.87|0.87|0.87|
>
> **(Continued on the next response)**

---

> > ### Author Response · Authors · 2022-11-17
> > **To Reviewer 7vyQ (2/2)**
> >
> > For "B1(r)", it can be seen that the larger the "$r$" (1.0/0.5/0.1) is, the better the results are (except for T-20), which is obviously because it can utilize more data. What is interesting is that even "B1(1.0)", which uses all the previous data, can be worse than MTL in some cases (e.g., C-10, T-10, and T-20). We believe this reason is that sequentially establishing different decision boundaries for multiple tasks may sacrifice the effectiveness of learning. Although SPG is weaker than "B1(1.0)" in dissimilar tasks, it outperforms “B1(0.5)”, which indicates SPG is more effective at least than storing one half of the previous data. Note that SPG does not store any previous data. For similar tasks, B1(0.5) and B(0.1) are weaker than SPG. B1(1.0) has the same results as SPG.
> >
> > For "B2", it enjoys forward knowledge transfer, but since B2 is not a continual learning method, it has no catastrophic forgetting (CF) issue. It is thus not surprising that B2 outperforms SPG on the first two dissimilar tasks (C-10 and C-20). But SPG shows competitive performance with B2 even under CF of continual learning for T-10 and T-20, and significantly outperforms B2 on I-100. To have a better understanding of the interesting phenomenon, we conducted additional experiments with more tasks, T-50, T-100, I-50, and I-200. We got the following results,
> >
> > |Model|T-50|T-100|I-50|I-100|I-200|
> > |--|--|--|--|--|--|
> > |ONE|0.71|0.85|0.40|0.49|0.60|
> > |B2|0.72|0.84|0.38|0.48|0.62|
> > |**SPG**|0.71|0.84|0.45|0.55|0.66|
> >
> > We can see that B2 is not better than SPG and in fact for the last three, it is actually significantly poorer than SPG. We believe the reason is as follows. The initialization of parameters affects the performance of learning of a task, and a model that has learned a task does not always provide good initialized parameters for the next task. For the ImageNet data, which is used for I-50, I-100, and I-200, it is a more difficult dataset. In this case, the model parameters get optimized more specifically for each task, and the learning of the next task in B2 may start with parameters that are far from a good initialization, which leads to poorer results than SPG. On the other hand, since SPG preserves knowledge from all previous tasks, the accumulated knowledge in the network enables the model to utilize a good set of parameters to learn the new task (see below for similar tasks as well) to produce better results, which is indicated by its strong transfer compared to ONE and also as described in Table 3 in the paper.
> >
> > SPG is also consistently better than B2 for similar tasks (see FC-10 to FE-20). This is because of the limited knowledge transfer in B2. Let the current task learned by B2 be $t$. B2 can only transfer knowledge from the $t-1$ model to task $t$. Although some knowledge of earlier tasks before $t-1$ may still be in the model for $t-1$, it is largely forgotten because B2 has no mechanism to handle forgetting. Our SPG can preserve the task specific knowledge of each task and thus is able to leverage the knowledge from more and suitable tasks to learn task $t$ better.

---

> > > ### Comment · Reviewer_7vyQ · 2022-11-18
> > > **response to response**
> > >
> > > | We have seen a lot of applications of task-incremental learning (TIL). We humans learn a lot of separate tasks all the time, e.g., (1) classifying different kinds of birds and (2) classifying different types of animals (notice the overlap of the concepts, which is not suitable for class-incremental learning).
> > > We do not operate in a world where an oracle consistently gives us a \emph{task ID} is what I meant by TIL being overly artificial. We also don't constrain ourselves from using replay to refresh our knowledge. Applications for TIL as it is formulated remain nonexistent in my opinion.
> > >
> > > | We believe that your suggested baselines are not continual learning (CL) methods. For the first baseline, when the ratio is 1, we need to save all the data, which is like running multitask learning at each task.
> > > Continual learning is a problem, we shouldn't restrict ourselves to methods that seem like continual learning approaches: we should simply solve the problem as efficiently as possible.
> > >
> > > |  The computational cost is huge (O(n^2), where (n) is the number of tasks in a dataset.
> > > Absolutely not. The author has assumed to approximate a balanced distribution on all tasks forever. This is not required at all.
> > >
> > > | For the second baseline, it is not a CL setting because it saves a model for each task and the model is trained with the parameters learned from all previous tasks as the initialization.
> > > What? Why is this not a CL setting?
> > >
> > > | We can see that B2 is not better than SPG and in fact for the last three, it is actually significantly poorer than SPG. We believe the reason is as follows. The initialization of parameters affects the performance of learning of a task, and a model that has learned a task does not always provide good initialized parameters for the next task.
> > >
> > > Perfect, this is an important TIL baseline that does indeed suffer from the aforementioned weakness. Nevertheless, it is important to beat it.
> > >
> > > I hope the results will be added to the text.
> > > I also hope the authors will stop dismissing baselines because they do not obey their continual learning requirements.
> > > I'm updating my score to a 6

---

> > > > ### Author Response · Authors · 2022-11-18
> > > > **To Reviewer 7vyQ (1/1)**
> > > >
> > > > > We do not operate in a world where an oracle consistently gives us a \emph{task ID} is what I meant by TIL being overly artificial. We also don't constrain ourselves from using replay to refresh our knowledge. Applications for TIL as it is formulated remain nonexistent in my opinion.
> > > >
> > > > We agree that without using task ID is better. But when tasks have overlapping concepts, task related information (hint, context or task ID) will be useful. For example, one task model classifies different breeds of dogs and another task model classifies only dogs and cats (two classes). If the test sample is a specific dog, without knowing which task to apply, the system will not know what result to present to the user.
> > > >
> > > >
> > > >
> > > > > Continual learning is a problem, we shouldn't restrict ourselves to methods that seem like continual learning approaches: we should simply solve the problem as efficiently as possible.
> > > >
> > > > We agree. In this paper, we focus on the continual learning setting.
> > > >
> > > >
> > > > > Absolutely not. The author has assumed to approximate a balanced distribution on all tasks forever. This is not required at all.
> > > >
> > > > Yes, when the data is small the computation will be less.
> > > >
> > > >
> > > >
> > > > > What? Why is this not a CL setting?
> > > >
> > > > Yes, it can be a CL setting. It is just that saving a large model for each task has a scalability issue when the number of tasks is very large, so most CL methods share parameters across tasks.
> > > >
> > > >
> > > >
> > > > > Perfect, this is an important TIL baseline that does indeed suffer from the aforementioned weakness. Nevertheless, it is important to beat it.
> > > >
> > > > Thanks for your suggestion. This analysis has indeed revealed an interesting phenomenon. We would like to continue the investigation on this in our future work.
> > > >
> > > >
> > > >
> > > > > I hope the results will be added to the text. I also hope the authors will stop dismissing baselines because they do not obey their continual learning requirements. I'm updating my score to a 6.
> > > >
> > > > We have added the results in Appendix A.7, and referred to it in the main text (see page 6), which is highlighted in blue. Thanks, we will not dismiss them.
> > > > We very much appreciate the raise of your score.

---

### Official Review · Reviewer_s4uK · 2022-10-25

**Confidence:** 4
**Correctness:** 2
**Technical Novelty And Significance:** 2
**Empirical Novelty And Significance:** 2
**Recommendation:** 3

**Clarity, Quality, Novelty And Reproducibility:**

The clarity of the work is acceptable.
The overall quality of the work does not meet the standard of ICLR.
The novelty is incremental.
The reproducibility is questionable.

**Strength And Weaknesses:**

The main strength of the paper is that it is relatively easy to follow and understand the high-level idea.

Weaknesses:
- The topic of task-incremental learning is not particular interesting, given its proven simplicity in prior work.
- The core idea of the work is incremental: using importance mask to condition the model is common in architecture-based methods.
- The empirical evaluation is questionable. For example, in Table 2, all methods in the table has very small standard deviation. I do not think it corresponds with reported results in prior work.

**Summary Of The Paper:**

The paper presents a new method (SPG) for task-incremental learning. SPG uses soft-masks to condition the full model for each task. Experiment results on benchmark datasets demonstrate the effectiveness of the proposed method.

**Summary Of The Review:**

The overall quality of the work does not meet the standard of ICLR, given the limited novelty and simplicity of the topic. Substantial amount of work is required to improve the paper. Therefore, I would recommend rejection.

---

> ### Author Response · Authors · 2022-11-17
> **To Reviewer s4uK (1/1)**
>
> We appreciate your review. We answer your concerns as follows.
>
> [Response 1]
> > The topic of task-incremental learning is not particular interesting, given its proven simplicity in prior work.
>
> It is true that task-incremental learning (TIL) is not hard in terms of overcoming catastrophic forgetting. However, knowledge transfer and the limited capacity of the existing methods are challenging issues and are far from solved for TIL. Knowledge transfer is one of the two major objectives of continual learning and is very important in applications, but not much research attention has been paid to it. This paper deals with forgetting, limited capacity, and knowledge transfer in TIL. To our knowledge, none of the existing methods has attempted all three. Knowledge transfer and limited capacity problems are non-trivial. As a community, we should not discriminate against TIL as it has its own wide ranges of applications. We humans learn a lot of separate tasks all the time, e.g., (1) classifying different kinds of birds and (2) classifying different types of animals (notice the overlap of concepts, which is not suitable for class-incremental learning).
>
>
>
>
>
> [Response 2]
> > The core idea of the work is incremental: using importance mask to condition the model is common in architecture-based methods.
>
> Although existing methods (HAT and SupSup are representatives) use importance masks to condition the model, their masks are binary and have capacity issues and also have difficulty to transfer knowledge across tasks (see the introduction section). We use soft-masks, which are very well suited to deal with these problems. To our knowledge, soft-masks have not been used so far. Although parameter importance is commonly used in regularization based methods for CL, the fundamental difference between our approach and regularization-based methods is that in regularization-based methods, the parameter importance is used in a penalty term in the loss to penalize parameter updates, which is not very effective as we have discussed in the second contribution (2) in page 2. Our experiments also show soft-masking is much better than the regularization-based approach. The reason is that our approach has strict control over the update of each individual parameter, but regularization-based methods do not, as the penalty term does not control each parameter individually (see (2) in page 2).
>
>
>
> [Response 3]
> > The empirical evaluation is questionable. For example, in Table 2, all methods in the table has very small standard deviation. I do not think it corresponds with reported results in prior work.
>
> We have checked the original papers of the baselines, but most of them (e.g., PGN, PathNet, HAT, CAT, UCL, SI, and EWC) did not report the standard deviation (std). TAG (https://arxiv.org/pdf/2105.05155.pdf) reported that TAG/EWC/MTL has 0.29%/3.44%/0.58% std for C-20 while TAG/EWC/MTL in our experiments has 0.9%/2.7%/0.4%, respectively. As you can see, our std values are not so different from theirs. A-GEM (https://arxiv.org/pdf/1812.00420.pdf) reported the std of less than 1% and ours are actually larger. SupSup (https://arxiv.org/pdf/2006.14769.pdf) also reports its std to be less than 1% and ours are less or equal to 1% as well. Thus, we do not see that our experiments are questionable. Please note that all the methods are evaluated in the same way.

---

> > ### Author Response · Authors · 2022-11-18
> > **Additional comment to our [Response 3]**
> >
> > It just comes to us about the reason why you might have thought that our standard deviation (std) values are too small. We reported the accuracy and std values in the decimal form, e.g., our 0.7 means 70%, but most existing papers report their results in percentage (without the % sign), e.g.,70.0 (70%). So, our std of 0.01 is 1.0 in their papers, which might be why you thought that our std values are very small at a glance.

---

> > ### Comment · Reviewer_s4uK · 2022-11-19
> > **Additional Response**
> >
> > Thank the authors for the response, some of my concerns are addressed. However, I still have several concerns:
> >
> > 1. The limitation of task-incremental learning is *assuming known task ID at test time*. Moreover, please clarify does any benchmarks in the experiments corresponds to this concept overlap case? As far as I am concerned, split versions of CIFAR and ImageNet are class-incremental learning benchmarks with given test time task ID. Therefore, I still think the scope of the paper is somewhat limited.
> >
> > 2. Please double check the claim "soft-masks have not been used so far". For example, the faster learner in DualNet (https://arxiv.org/pdf/2110.00175.pdf) can be regarded as soft-masks.

---

> > > ### Author Response · Authors · 2022-11-21
> > > **To Reviewer s4uK (1/2)**
> > >
> > > > The limitation of task-incremental learning is assuming known task ID at test time. Moreover, please clarify does any benchmarks in the experiments corresponds to this concept overlap case? As far as I am concerned, split versions of CIFAR and ImageNet are class-incremental learning benchmarks with given test time task ID. Therefore, I still think the scope of the paper is somewhat limited.
> > >
> > > Providing task related information (e.g., task ID) is also the same for humans. For example, assume a person has learned many tasks related to images. If you give him an image without telling him what you want him to do about it (task information), he will not know whether you want him to classify the image into a broad category (e.g., dog among other kinds of animals) or a fine-grained category (a specific breed of dog), to write a caption for it, or even to segment the image. Task-incremental learning (TIL) is basically a continual learning setting for this situation. Thus, we strongly believe that different settings (class-incremental learning (no task ID) and TIL) should be treated equally as they are all very useful in practice.
> > >
> > > We did not have experiments using overlapping concepts as we follow existing task incremental learning (TIL) papers and they also do not have such experiments. But thanks for the suggestion. We have created two overlapping settings and conducted experiments on them.
> > >
> > > Exp1: A dataset is split into multiple tasks and the tasks have overlapping classes. Here, we split CIFAR100 into 10 tasks (i.e., task 1: classes 0-14, task 2: classes 10-24, task 3: classes 20-34, ..., task 9: classes 80-94, task 10: classes 90-99).
> > >
> > > Exp2: As CIFAR100 has superclasses (see https://www.cs.toronto.edu/~kriz/cifar.html in detail), we create tasks to perform different levels of classification, e.g., some tasks predict "fine-grained" labels of objects in the same superclass, and others predict "coarse" labels of objects across different superclasses. For example, task 1 contains the set of fine-grained classes "beaver", "dolphin", "otter", "seal", and "whale" from the superclass "aquatic mammals", and task 2 contains images of “dolphin” (as "aquatic mammals"), "fish", "insects", "flowers", and "trees", where all the labels are coarse or superclasses. In this case, task 1 and task 2 have the overlapping concept of dolphins, and the model needs task ID to know whether we want to predict a fine-grained or coarse label. For this case, we split CIFAR100 into 40 tasks.
> > >
> > > The results are presented below. We compare SPG with 5 best performing baselines in addition to ONE and NCL. Note that multitask learning (MTL) in this case does not make sense due to overlapping concepts/classes.
> > >
> > > ||Exp1|Exp2|
> > > |--|--|--|
> > > |(MTL)|-|-|
> > > |(ONE)|$0.58 \pm 0.008$|$0.60 \pm 0.006$|
> > > |NCL|$0.46 \pm 0.019$|$0.31 \pm 0.008$|
> > > |PathNet|$0.58 \pm 0.008$|$0.58 \pm 0.007$|
> > > |HAT|$0.56 \pm 0.010$|$0.55 \pm 0.006$|
> > > |SupSup|$0.51 \pm 0.004$|$0.53 \pm 0.010$|
> > > |UCL|$0.57 \pm 0.012$|$0.55 \pm 0.004$|
> > > |SI|$0.57 \pm 0.004$|$0.56 \pm 0.008$|
> > > |**SPG**|$0.60 \pm 0.003$|$0.60 \pm 0.004$|
> > >
> > > We can see that SPG enjoys its effectiveness in these overlapping concept experiments and outperforms all baselines. Our SPG is also on par or better than ONE (which trains a separate network for each task). We will include these experiments and results in the revised paper. Thanks.

---

> > > > ### Author Response · Authors · 2022-11-21
> > > > **To Reviewer s4uK (2/2)**
> > > >
> > > > > Please double check the claim "soft-masks have not been used so far". For example, the faster learner in DualNet (https://arxiv.org/pdf/2110.00175.pdf) can be regarded as soft-masks.
> > > >
> > > > We have done extensive literature search and have not found any existing approach similar to our soft-masking approach. DualNet also does not do soft-masking. It consists of a fast learner and a slow learner. The slow learner does feature learning via self-supervised learning (SSL). We quote "The slow learner is a standard backbone network $\phi$ that is trained to optimize an SSL loss." The faster learner learns with the labeled data. We quote "Given a labeled sample {$x$, $y$}, the fast learner’s goal is utilizing the slow learner’s representation to quickly learn this sample via an adaptation mechanism." The adaptation mechanism works as follows. The output of layer $l$ of the fast learner is the output of layer $l$ in the slow learner element-wise multiplied with a factor computed from the output of $l-1$ layer of the fast learner (see Eq. 5 and Eq. 6 in the DualNet paper). As we can see, this is done in the forward pass and is very different from our soft-masking. What soft-masking does in SPG is to suppress the update of each parameter’s gradient in the backward pass to prevent forgetting (see Eq. 5 in our paper, where $g_i$ , a vector, represents the gradients of parameters in layer $i$ before soft-masking, and $g_i^\prime$ means the modified gradients that are actually used in the optimization step). Soft-masking in SPG thus directly controls the update of each parameter based on its importance score computed based on the gradient, which DualNet does not do. Note that the symbol $g_{\theta,l}$ in DualNet is similar to our symbol for gradients $g_i^\prime$, which may be the confusion, but $g_{\theta,l}$ in DualNet denotes the $l$-th layer’s output from the fast network $\theta$. So the two symbols have completely different meanings.
> > > >
> > > > Hope we have cleared your doubt. If you have additional questions, please let us know and we are very happy to answer.

---

> > > > > ### Author Response · Authors · 2022-12-07
> > > > > **A gentle reminder**
> > > > >
> > > > > Dear Reviewer s4uK,
> > > > >
> > > > > As the end of the discussion period is coming (Dec. 12), we are wondering if your concerns have been addressed by our responses. If you have additional concerns or questions please let us know. We are happy to answer them.
> > > > >
> > > > > Thank you,
> > > > >
> > > > > Authors

---

### Official Review · Reviewer_5rH3 · 2022-10-26

**Confidence:** 4
**Correctness:** 4
**Technical Novelty And Significance:** 3
**Empirical Novelty And Significance:** 1
**Recommendation:** 6

**Clarity, Quality, Novelty And Reproducibility:**

- The overall performance seems meaningful, but it is not well explained why the proposed method works better than others.
- Are there some results comparing 'soft'-masking with 'hard'-masking?
- Recalculating the results by measuring the 'improvements from NCL performance', it will give better understanding of overcoming CF.

**Strength And Weaknesses:**

(+) it simply uses gradient masks that can be easily computed by using gradient values.

(+) the importance values can also be used in EWC framework, and this is computationally cheaper than using the Fisher information.

(+) masking is only applied during the learning phase, and it is not required during the inference time.

(-) the proposed method shows competitive performance compered to other methods, but it is not well-studied why it works better than others.

(-) it is unclear why the loss $\text{Sum}(\mathcal{M}_\tau(X_t))$ for the previous task $\tau < t$ is useful to compute the importance value.



**Summary Of The Paper:**

The goal of proposed method is to overcome the catastrophic forgetting in continual setting, while promoting the forward knowledge transfer, and it proposes a simple soft-masking approach applied to the gradients during back-propagation.

**Summary Of The Review:**

- The method is simpler than other methods and the overall performance seems meaningful, but it is not well explained why the proposed method works better than others.

---

> ### Author Response · Authors · 2022-11-17
> **To Reviewer 5rH3 (1/2)**
>
> We appreciate your review. We answer your concerns as follows.
>
> [Response 1]
> > (-) the proposed method shows competitive performance compered to other methods, but it is not well-studied why it works better than others.
>
> As we explained in the introduction, ideally, TIL methods should (a) overcome forgetting, (b) have capability to keep learning new tasks, and (c) transfer knowledge across tasks. Perhaps the two most effective TIL methods are HAT and SupSup as both of them have no forgetting (a). We show that HAT has the capacity problem (b) and SupSup cannot transfer knowledge at all (c). The key issue is that they use masks to protect or block previously learned parameters. Our method keeps (a) and has a much higher capacity (b) and also automatically transfers knowledge (c). As we have discussed in the later part in the introduction section, it is because our soft-masking method SPG does not fully block any parameters, which naturally makes SPG better in terms of (b) and (c). (a) is also achieved due to the soft mask and its knowledge transfer ability (see the second paragraph in Section 4.2). In fact, using parameter importance is already common in regularization-based approaches for overcoming forgetting (a). However, the fundamental difference between our approach and regularization-based methods is that in regularization-based methods, the parameter importance is used in a penalty term in the loss to penalize parameter updates, which is not very effective as we have discussed in the second contribution (2) in page 2. Our experiments also show soft-masking is much better than the regularization-based approach. The reason is that SPG has strict control over the update of each individual parameter, but regularization-based methods do not, as the penalty term does not control each parameter individually (see (2) in page 2).
>
>
> [Response 2]
> > (-) it is unclear why the loss $\mathrm{Sum}(M_\tau(X_t))$ for the previous task $\tau < t$ is useful to compute the importance value.
>
> We use this loss in cross-head importance (CHI) for previous tasks to know how much each parameter can affect previous tasks. Although the importance of each parameter has already been computed just after learning a previous task $\tau$ and the parameter is updated based on soft-masking in learning the new task $t$, it might have been changed to some degree. Since the accumulation of many small changes on all parameters may affect the dynamics of the network (i.e., relative importance of each parameter) significantly in total, CHI is introduced to reflect the importance again in the current network state after learning task $t$. Given no previous data, we substitute the current task’s data as unlabeled data for previous tasks. While the current task’s data is different from previous tasks, we hypothesize that this can be used as hints to get to know the relative importance of parameters in the current network state. Our hypothesis has been tested through the ablation study in Table 6 by showing that CHI is effective in mitigating forgetting.
>
>
> [Response 3]
> > The overall performance seems meaningful, but it is not well explained why the proposed method works better than others.
>
> Please see our [Response 1]. We explained why our method works better in the introduction section.

---

> > ### Author Response · Authors · 2022-11-17
> > **To Reviewer 5rH3 (2/2)**
> >
> > [Response 4]
> > > Are there some results comparing 'soft'-masking with 'hard'-masking?
> >
> > HAT uses hard-masking (on neurons) and we have shown that our SPG (soft-masking parameters) outperformed HAT significantly. SupSup hard-masks parameters. SPG also outperforms it.
> >
> > Although SPG directly uses importance scores (0 to 1) to mask parameters and does not have a threshold for hard-masking (0 or 1), we create a hard mask version of SPG, called "HPG" (i.e., Hard-masking, not soft-masking) by applying a threshold to convert an importance value (0 to 1) to a binary value (0 or 1). For example, if the threshold is 0.5, HPG treats importance larger than 0.5 as 1 (blocking), otherwise 0 (not blocking). We search for the best threshold for each dataset from 0.1/0.3/0.5/0.7/0.9, and report the best accuracy result in the following table.
> >
> > (C-10 to I-100 are dissimilar datasets, while FC-10 to FE-20 are similar datasets)
> >
> > |Model|C-10|C-20|T-10|T-20|I-100|FC-10|FC-20|FE-10|FE-20|
> > |--|--|--|--|--|--|--|--|--|--|
> > |(MTL)|0.76|0.78|0.53|0.60|0.65|0.88|0.88|0.86|0.87|
> > |(ONE)|0.67|0.76|0.44|0.54|0.49|0.75|0.79|0.81|0.80|
> > |NCL|0.51|0.54|0.37|0.41|0.31|0.84|0.84|0.86|0.86|
> > |HAT|0.63|0.72|0.46|0.52|0.45|0.79|0.82|0.84|0.85|
> > |SupSup|0.66|0.76|0.44|0.54|0.49|0.71|0.72|0.81|0.80|
> > |HPG|0.63|0.73|0.45|0.53|0.49|0.87|0.86|0.87|0.87|
> > |**SPG**|0.66|0.75|0.47|0.57|0.55|0.86|0.87|0.87|0.87|
> >
> >
> >
> > The results still show SPG is superior. For dissimilar tasks (column 2-6), HPG that does hard-masking is significantly worse than SPG, which means that it is difficult to manually control the threshold and it is more effective to use the importance scores as soft-masks (as in SPG). On the other hand, both HPG and SPG work similarly for similar tasks (the last 3 columns), which is reasonable as for similar tasks, completely blocking parameters is not as harmful as for dissimilar tasks because similar tasks share a lot of knowledge or parameters.
> >
> >
> >
> >
> > [Response 5]
> > > Recalculating the results by measuring the 'improvements from NCL performance', it will give better understanding of overcoming CF.
> >
> > Thanks for the suggestion. We are not completely sure what you meant by recalculating as Tables 2, 3, and 4 in the main text have the results for each baseline including NCL. So it should be easy to see the difference between each system and NCL. However, since you said "it will give better understanding of overcoming CF," we think what you meant was to compare the final accuracy of each system and that of NCL, which does give a good indication on how each system fares in terms of forgetting compared to NCL. We added the results in Table 11 in Appendix 6, which shows that our SPG does much better than all baselines.

---

### Decision · Program_Chairs · 2023-01-20

**Decision:**

Reject

**Justification For Why Not Higher Score:**

Not so novel and some experiment results are questionable.

**Justification For Why Not Lower Score:**

N/A

**Metareview: Summary, Strengths And Weaknesses:**

Authors have proposed a new SPG method for task-incremental learning. Specifically, SPG uses soft-masks to condition the full model for each task. It conducts experiments to demonstrate the effectiveness of the proposed method on benchmark datasets .

The novelty of this paper is limited as using importance mask to condition the model is common in architecture-based methods.

**Summary Of Ac-Reviewer Meeting:**

No need to do the meeting as the reviewers do not like the paper